# Improving the Potential Accuracy and Usability of EURO-CORDEX Estimates of Future Rainfall Climate using Frequentist Model Averaging

Stephen Jewson[1], Giuliana Barbato[2], Paola Mercogliano[2], Jaroslav Mysiak[2], Maximiliano Sassi[3]

[1]Independent Researcher, London, UK

[2]Euro-Mediterranean Center on Climate Change (CMCC) Foundation, Via Augusto Imperatore, 16, 73100, Lecce, Italy

[3]Risk Management Solutions Ltd, EC3R 7AG, London, UK

*Correspondence to*: Stephen Jewson (stephen.jewson@gmail.com)

**Abstract.** Probabilities of future climate states can be estimated by fitting distributions to the members of an ensemble of climate model projections. The change in the ensemble mean can be used as an estimate of the change in the mean of the real climate. However, the level of sampling uncertainty around the change in the ensemble mean varies from case to case and in some cases is large. We compare two model averaging methods that take the uncertainty in the change in the ensemble mean into account in the distribution fitting process. They both involve fitting distributions to the ensemble using an uncertainty-adjusted value for the ensemble mean in an attempt to increase predictive skill relative to using the unadjusted ensemble mean. We use the two methods to make projections of future rainfall based on a large dataset of high resolution EURO-CORDEX simulations for different seasons, rainfall variables, RCPs and points in time. Cross-validation within the ensemble using both point and probabilistic validation methods shows that in most cases predictions based on the adjusted ensemble means show higher potential accuracy than those based on the unadjusted ensemble mean. They also perform better than predictions based on conventional Akaike model averaging and statistical testing. The adjustments to the ensemble mean vary continuously between situations that are statistically significant and those that are not. Of the two methods we test, one is very simple, and the other is more complex and involves averaging using a Bayesian posterior. The simpler method performs nearly as well as the more complex method.

## 1 Introduction

Estimates of the future climate state are often created using climate projection ensembles. Examples of such ensembles include the CMIP5 project (Taylor, et al., 2012), the CMIP6 project (Eyring, et al., 2016) and the EURO-CORDEX project

(Jacob, Petersen, & authors, 2014). If required, distributions can be fitted to these ensembles to produce probabilistic predictions. The probabilities in these predictions are conditional probabilities and depend on the assumptions behind the climate model projections, such as the choice of RCP (Moss, et al., 2010; Meinshausen, et al., 2011), and the choice of models and model resolution. Converting climate projection ensembles to probabilities in this way is helpful for those applications in which the smoothing, interpolation and extrapolation provided by a fitted distribution is beneficial. It is also helpful for those applications for which the impact models can ingest probabilities more easily than they can ingest individual ensemble members. An example of a class of impact models that, in many cases, possess both these characteristics would be the catastrophe models used in the insurance industry. Catastrophe models quantify climate risk using simulated natural catastrophes embedded in many tens of thousands of simulated versions of one year (Friedman (1972), Kaczmarska, et al. (2018), Sassi, et al. (2019)). Methodologies have been developed by which these catastrophe model ensembles can be adjusted to include climate change, based on probabilities derived from climate projections (Jewson, et al., 2019).

A number of studies have investigated the post-processing of climate model ensembles. These studies have addressed issues such as estimation uncertainty (Deser, et al. (2010), Thompson, et al. (2015), Mezghani, et al. (2019)), how to break the uncertainty into components (Hawkins and Sutton, (2009), Yip, et al. (2011), Hingray and Said (2014)), how to identify forced signals given the uncertainty (Frankcombe, et al. (2015), Sippel, et al. (2019), Barnes, et al. (2019) and Wills, et al. (2020)), how quickly signals emerge from the noise given the uncertainty (Hawkins and Sutton (2012), Lehner, et al. (2017)), and how to apply weights and bias corrections (Knutti et al. (2010), Christensen et al. (2010), Buser et al. (2010), Deque et al. (2010), DelSole et al. (2013), Sanderson et al. (2015b), Knutti et al. (2017), Mearns et al. (2017), Chen et al. (2019)). In this article, we explore some of the implications of estimation uncertainty in climate model ensembles in more detail. We will consider the case in which distributions are fitted to climate model outputs, and in particular to changes in climate model output, rather than to absolute values. When fitting distributions to changes in climate model output, the change in the ensemble mean can be used as an estimate of the change in the mean of the real future climate. However, because climate model ensembles are finite in size, and different ensemble members give different results, the ensemble mean change suffers from estimation uncertainty when used in this way. Ensemble mean change estimation uncertainty varies by season, variable, projection, time and location. In the worst cases, the uncertainty may be larger than the change in the ensemble mean itself, and this makes the change in the ensemble mean, and distributions that have been fitted to the changes in the ensemble, potentially misleading and difficult to use. In these large uncertainty cases the change in the ensemble mean is dominated by the randomness of internal variability from the individual ensemble members, and it would be unfortunate if this randomness was allowed to influence adaptation decisions. A standard approach for managing this varying uncertainty in the change in the ensemble mean is to consider statistical significance of the changes (e.g., see shading of regions of statistical significance in climate reports such as the EEA report (European Environment Agency, 2017) or the IPCC 2014 report (Pachauri & Meyer, 2014)). Statistical significance testing involves calculating the signal-to-noise ratio (SNR) of the change in the ensemble mean, where the signal is the ensemble mean change, and the noise is the

standard error of the ensemble mean. The SNR is then compared with a threshold value. If the SNR is greater than the threshold then the signal is declared statistically significant (Wilks, 2011).

Use of statistical significance to filter climate projections in this way is often appropriate for visualisation and scientific
discovery. However, it is less appropriate as a post-processing method for climate model data that is intended for use in impact models. This is perhaps obvious, but it is useful to review why, as context and motivation for the introduction of alternative methods for managing ensemble uncertainty. To illustrate the shortcomings of statistical testing as a method for ensemble post-processing we consider a system which applies statistical testing and sets locations with non-significant values in the ensemble mean change to zero. The first problem with such a system is that analysis of the properties of
predictions made using statistical testing show that they have poor predictive skill. This is not surprising, since statistical testing was never designed as a methodology for creating predictions. The second problem is that statistical testing creates abrupt jumps of the climate change signal in space, between significant and non-significant regions, and between different RCPs and time points. These jumps are artefacts of the use of a method with a threshold. This may lead to situations in which one location is reported to be affected by climate change, and an adjacent location not, simply because the
significance level has shifted from e.g., 95.1% to 94.9%. From a practical perspective this may undermine the credibility of climate predictions in the perception of users, to whom no reasonable physical explanation can be given for such features of the projections. Finally, the almost universal use of a threshold p-value of 95% strongly emphasizes avoiding false positives (type I errors) but creates many false negatives (type II errors). Depending on the application, this may not be appropriate. Large numbers of false negatives is particularly a problem for risk modelling, since risk models should attempt to capture all
possibilities in some way, even if low significance.

How, then, should those who wish to make practical application of climate model ensembles deal with the issue of varying uncertainty in the changes implied by the ensemble, in cases where for many locations the uncertainty is large and the implied changes are dominated by randomness? This question might arise in any of the many applications of climate model output, such as agriculture, infrastructure management, investment decisions, and so on. We describe and compare three
Frequentist Model Averaging (FMA) procedures as possible answers to this question. Frequentist model averaging methods (Burnham & Anderson, 2002; Hjort & Claeskens, 2003; Claeskens & Hjort, 2008; Fletcher, 2019) are simple methods for combining outputs from different models in order to improve predictions. They are commonly used in economics (Hansen, 2007; Liu, 2014). Relative to Bayesian model averaging methods (Hoeting, Madigan, Raftery, & Volinsky, 1999) they have various pros and cons (Burnham & Anderson, 2002; Hjort & Claeskens, 2003; Claeskens & Hjort, 2008; Fletcher, 2019). For
our purposes, we consider the simplicity, transparency and ease of application of FMA as benefits. The averaging in our applications of FMA consists of averaging of the usual estimate for the mean change with an alternative estimate of the change which is set to zero. This has the effect of reducing the ensemble mean change towards zero. The averaging weights, which determine the size of the reduction, depend on the SNR and are designed to increase the accuracy of the prediction. They vary in space, following the spatial variations in SNR. In regions where the SNR is large these methods make no

material difference to the climate prediction. In regions where the SNR is small, the changes in the ensemble mean are reduced in such a way as to increase the accuracy of the predictions.

This approach can be considered as a continuous analog of statistical testing, in which rather than setting the change in the ensemble mean to either 100% or 0% of the original value, we allow a continuous reduction that can take any value between 100% and 0% depending on the SNR. As a result, the approach avoids the abrupt jumps created by statistical testing. In

summary, by reducing the randomness in the ensemble mean (relative to the unadjusted ensemble mean), increasing the accuracy of the predictions (relative to both the unadjusted ensemble mean and statistical testing), and avoiding the jumps introduced by statistical testing, the FMA predictions may make climate model output more appropriate for use in impact models i.e., more usable. The increases accuracy are, however, not guaranteed, and need to be verified using potential accuracy, as we describe below.

One of the three FMA methods we apply is a standard approach based on the Akaike Information Criterion (AIC) (Burnham & Anderson, 2002), which we will call AIC model averaging (AICMA). The other two methods are examples of Least Squares Model Averaging (LSMA) methods (Hansen, 2007), also known as minimum mean squared error model averaging methods (Charkhi, Claeskens, & Hansen, 2016), which are FMA methods that focus on minimizing the mean squared error. The two LSMA methods we consider both work by using a simple bias-variance trade-off argument to reduce the change

captured by the ensemble mean when it is uncertain. One of them is a standard method, and the other is a new method that we introduce. We will call both LSMA methods 'Plug-in Model Averaging' (PMA), since they involve the simple, and standard, approach of 'plugging-in' parameter estimates into a theoretical expression for the optimal averaging weights (Jewson & Penzer, 2006; Claeskens & Hjort, 2008; Liu, 2014; Charkhi, Claeskens, & Hansen, 2016). The first PMA procedure we describe uses a simple plug-in estimator, and we refer to this method as Simple PMA (SPMA). The second

procedure is novel and combines a plug-in estimator with integration over a Bayesian posterior, and we refer to this method as Bayesian PMA (BPMA).

We illustrate and test the AICMA, SPMA and BPMA methods using a large dataset of high-resolution EURO-CORDEX ensemble projections of rainfall over Europe. We consider four seasons, three rainfall variables, two RCPs and three future time periods, giving 72 cases in all. In section 2 we describe the EURO-CORDEX data we will use. In section 3 we describe

AICMA and both PMA procedures, and present some results based on simulated data which elucidate the relative performance of the different methods in different situations, for both point and probabilistic predictions. In section 4 we present results for one of the 72 cases in detail. We use cross-validation within the ensemble to evaluate the potential prediction skill of the FMA methods, again for both point and probabilistic predictions, and compare with the skill from using the unadjusted ensemble mean and statistical testing. In section 5 we present aggregate results for all 72 cases using

the same methods. In section 6 we summarize and conclude.

## 2 Data and Methodology

The data we use for our study is extracted from the data archive produced by the EURO-CORDEX program (Jacob, Petersen, & authors, 2014; Jacob, Teichmann, & authors, 2020), in which a number of different global climate model simulations were downscaled over Europe using regional models at 0.11-degree resolution (about 12km). We use data from 10 models, each of which is a different combination of a global climate model and a regional climate model. The models are listed in Table 1. Further information on EURO-CORDEX and the models is given in the guidance report (Benestad, et al., 2017).

| Model | Driving GCM | GCM Member | RCM |
|-------|-------------|------------|-----|
| M1 | CNRM-CM5 | r1i1p1 | ALADIN53 |
| M2 | IPSL-CM5A-MR | r1i1p1 | RCA4 |
| M3 | CNRM-CM5 | r1i1p1 | RCA4 |
| M4 | CNRM-CM5 | r1i1p1 | CCLM4-8-17 |
| M5 | EC-EARTH | r12i1p1 | CCLM4-8-17 |
| M6 | EC-EARTH | r12i1p1 | RACMO22E |
| M7 | EC-EARTH | r12i1p1 | RCA4 |
| M8 | EC-EARTH | r1i1p1 | RACMO22E |
| M9 | EC-EARTH | r3i1p1 | HIRHAM5 |
| M10 | IPSL-CM5A-MR | r1i1p1 | WRF331F |

Table 1: Models used in this study

We extract data for four meteorological seasons (DJF, MAM, JJA, SON), for three aspects of rainfall: changes in the total rainfall (RTOT), the 95th percentile of daily rainfall (R95) and the 99th percentile of daily rainfall (R99). We say 'rainfall' even though in some locations we may be including other kinds of precipitation. We consider two RCPs, RCP4.5 and RCP8.5, and four 30-year time-periods: 1981-2010, which serves as a baseline from which changes are calculated, and the three target periods of 2011-2040, 2041-2070 and 2071-2100. In total this gives 72 different cases (four seasons, three variables, two RCPs and three target time periods).

Figure 1 illustrates one of the 72 cases: changes in winter (DJF) values for RTOT, from RCP4.5, for the years 2011-2040. This example was chosen as the first in the database, rather than for any particular properties it may possess. Figure 1a shows the ensemble mean change $\hat{\mu}_c$ (the mean change calculated from the 10 models in the ensemble) and Fig. 1b shows the standard deviation of the change $\hat{\sigma}_c$ (the standard deviation of the changes calculated from the 10 models in the ensemble). Fig. 1c shows the estimated SNR $\hat{s}$ calculated from the ensemble mean change and the standard deviation of change using the expression $\hat{s} = n^{1/2}|\hat{\mu}_c| / \hat{\sigma}_c$, where the $n^{1/2}$ term in this equation converts the standard deviation of change (a measure of the spread of the changes across the ensemble) to the standard error of the ensemble mean change (a

measure of the uncertainty around the ensemble mean change). Finally, Fig. 1d shows the regions in which the changes in the ensemble mean are significant at the 95% level, assuming normally distributed changes. In Fig. 1a we see that the ensemble mean change varies considerably in space, with notable increases in RTOT in Ireland, Great Britain, and parts of France, Germany, Spain, Portugal and elsewhere. In Fig. 1b we see that the standard deviation of change also varies considerably with the largest values over Portugal, parts of Spain and the Alps. In Fig. 1c we see that the SNR shows that many of the changes in Ireland, Great Britain, France, Germany and further east have particularly high SNRs (greater than four) while the changes in many parts of Southern Europe (Portugal, Spain, Italy and Greece) have lower values (often much less than one). Accordingly, Fig. 1d shows that the changes are statistically significant throughout most of Ireland, Great Britain, France, Germany, and Eastern Europe, but are mostly not statistically significant in Southern Europe. The other 71 cases show similar levels of variability of these four fields, but with different spatial patterns.

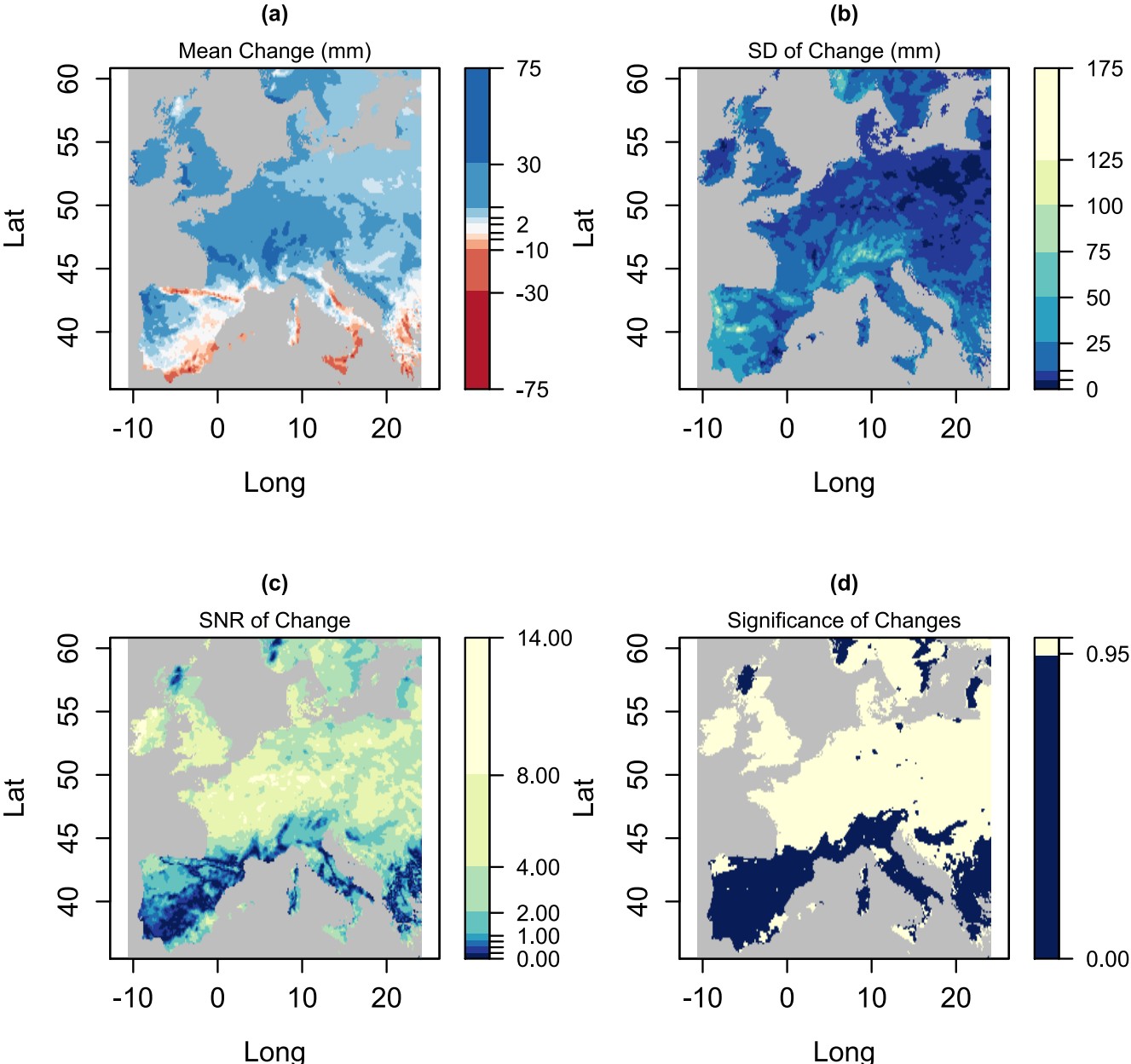

Figure 1: EURO-CORDEX projections for winter, for the change in total precipitation (RTOT) between the period 2011-2040 and the baseline 1981-2010, for RCP4.5. Panel (a) shows the ensemble mean change, panel (b) shows the ensemble standard deviation of change, panel (c) shows the signal-to-noise ratio (SNR) and panel (d) shows the regions in which the changes in the mean are significant at the 95% level (shaded in lighter colour).

Figure 2 shows spatial mean values of the SNR (where the spatial mean is over the entire domain shown in Fig. 1) for all 72 cases. Each black circle is a spatial mean value of the SNR for one case, and each of the four panels in Fig. 2 shows the same 72 black circles but divided into sub-categories in different ways. The horizontal lines are the averages over the black circles in each sub-category. Figure 2a sub-divides by season: we see that there is a clear gradient from winter (DJF), which shows

the highest values of the spatial mean SNR, to autumn (SON) which shows the lowest values of spatial mean SNR. Fig. 2b sub-divides by rainfall variable: in this case there is no obvious impact on the SNR values. Fig. 2c sub-divides by RCP. RCP8.5 shows higher SNR values, as we might expect, since in the later years RCP8.5 is based on larger changes in external forcing. Fig. 2d sub-divides by time-period: there is a strong gradient in SNR from the first of the three time-periods to the last. This is also as expected since both RCP scenarios are based on increasing external forcing with time. We would expect

these varying SNRs to influence the results from the FMA methods. This will be explored in the results we present below.

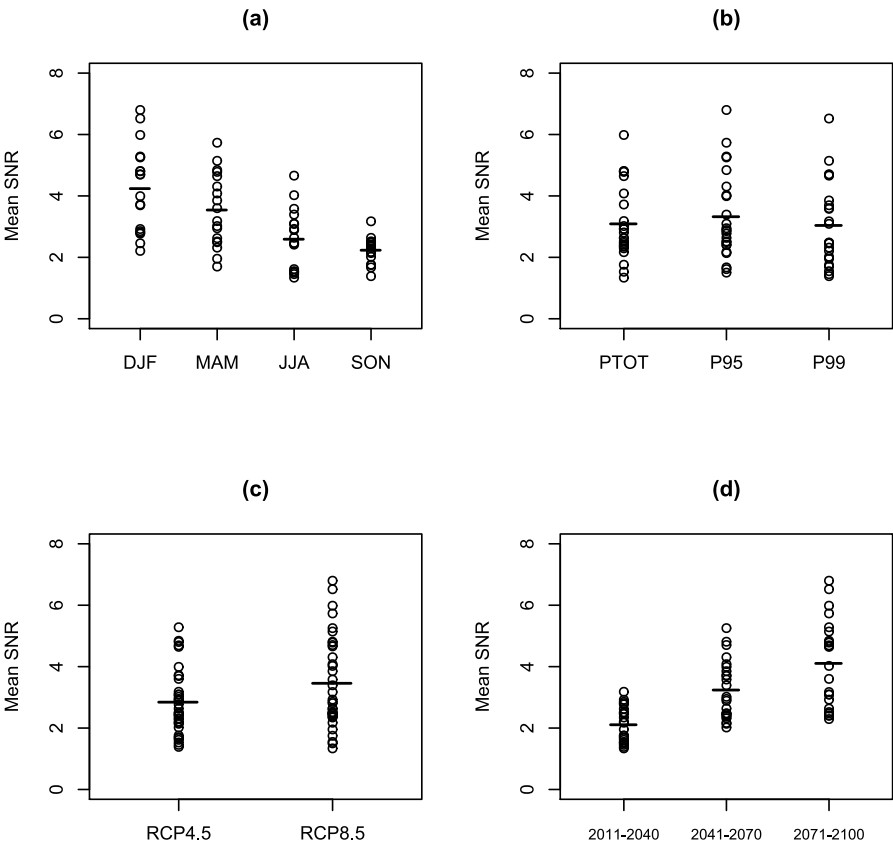

Figure 2: Each panel shows 72 values of the spatial average SNR (black circles) derived from each of the 72 EURO-CORDEX climate change projections described in the text, along with means within each subset (horizontal lines). Panel (a)

shows the 72 values as a function of season, panel (b) shows them as a function of rainfall variable, panel (c) shows them as a function of RCP and panel (d) shows them as a function of time period.

## 3 Model Averaging Methodologies

The model averaging methodologies we apply are used to average together uncertain projections of change with projections of no change, in such a way as to try and improve predictive skill. The AICMA method is a standard text-book method (Burnham & Anderson, 2002; Claeskens & Hjort, 2008). The weights are determined from the AICc score, which involves a small correction relative to the standard AIC score. The method attempts to minimise the difference between the real and predicted distributions, as measured using the Kullback-Leibler divergence. The PMA methods are based on a standard bias-variance trade-off argument, and the derivations of the methods follow standard mathematical arguments and proceed as follows.

### 3.1 Assumptions

For each location within each of the 72 cases, we first make some assumptions about the variability of the climate model results, the variability of future reality, and the relationship between the climate model ensemble and future reality. All quantities are considered as changes from the 1981-2010 baseline. We assume that the actual future value is a sample from a distribution with unknown mean $\mu$ and variance $\sigma^2$. We assume that the climate model values are independent samples from a distribution with unknown mean $\mu_c$ and variance $\sigma_c^2$. For the BPMA method we will additionally assume that these distributions are normal distributions. With regards to the assumption of independence of samples, this is an approximation, since the models are not entirely independent. Issues related to model dependence and independence have been discussed in various papers (see the citations in the introduction) but it is still unclear whether attempting to correct for dependence is beneficial or not, and so we do not. In terms of how the climate models and reality relate to each other, we assume that the climate model ensemble is realistic in the sense that it captures the real distribution of uncertainty, and so the mean and variance parameters agree, giving $\mu_c = \mu$ and $\sigma_c^2 = \sigma^2$. This is a "perfect ensemble" assumption. This is not likely to be strictly correct, and real climate model ensembles do contain errors and biases, but is a useful working assumption. We will write the future climate state as $y$, and the ensemble mean, estimated in the usual way from the ensemble, as $\hat{\mu}_c$. Since the usual estimator for the mean is unbiased, we can then say:

$$E(\hat{\mu}_c) = \mu_c = \mu = E(y) \tag{1}$$

If we write the ensemble variance, estimated using the usual unbiased estimator, as $\hat{\sigma}_c^2$, then we can say:

$$E(\hat{\sigma}_c^2) = \sigma_c^2 = \sigma^2 = V(y) \tag{2}$$

Uncertainty around the estimate of the ensemble mean is given by:

$$V(\hat{\mu}_c) = \frac{\sigma_c^2}{n} = \frac{\sigma^2}{n} \approx \frac{\hat{\sigma}_c^2}{n} \qquad (3)$$

### 3.2 The Simple Plug-in Model Averaging (SPMA) Methodology

The SPMA method we use is adapted from a method used in commercial applied meteorology, where the principles of bias-variance trade-off were used to derive better methods for fitting trends to observed temperature data for the pricing of weather derivatives (Jewson & Penzer, 2006). Similar methods have been discussed in the statistics and economics literature (Copas, 1983; Claeskens & Hjort, 2008; Charkhi, Claeskens, & Hansen, 2016). The adaptation and application of the method to ensemble climate predictions is described in a non-peer-reviewed technical report (Jewson & Hawkins, 2009a), but was

not tested extensively, and that report does not attempt to answer the question of whether the method really works in terms of improving predictions. The present study is, we believe, the first attempt at large-scale testing of any kind of FMA method using real climate predictions, and such testing is essential to evaluate whether the methods really are likely to improve predictions in practice.

In the SPMA method we make a new prediction of future climate in which we adjust the ensemble mean change using a

220 multiplicative factor $k$. $k$ is an averaging weight such that the weight on the ensemble mean is $k$ and the weight on a change of zero is $1 - k$. Combining different predictions using weights in this way is a standard method common to all model averaging schemes. We write the new prediction $\hat{y}$ as:

$$\hat{y} = k\,\hat{\mu}_c \qquad (4)$$

where the factor $k$, for which we derive an expression below, varies from 0 to 1 as a function of all the parameters of the

225 prediction: season, variable, RCP, time-period and spatial location. The intuitive idea behind this prediction is that if in one location the SNR in the ensemble is large, and hence the ensemble mean change prediction $\hat{\mu}_c$ is statistically significant, then it makes sense to use the ensemble mean more or less as is, and $k$ should be close to 1. On the other hand, if the SNR is small, and hence the change in the ensemble mean is far from statistically significant, then perhaps it is better to use a $k$ value closer to zero. Statistical testing sets $k$ to either 1 or 0 depending on whether the change is significant or not: the

230 SPMA method (and the BPMA method described later) allow it to vary continuously from 1 to 0.

The ensemble mean is the unique value that minimises MSE within the ensemble. However, when considering applications of ensembles, it is generally more appropriate to consider out of sample, or predictive, MSE (PMSE). We can calculate the statistical properties of the prediction errors for the prediction $\hat{y}$, and the PMSE, as follows:

$$\text{prediction error} = e = y - \hat{y} = y - k\,\hat{\mu}_c \qquad (5)$$

$$\text{bias} = E(e) = E(y - k\,\hat{\mu}_c) = E(y) - E(k\,\hat{\mu}_c) = \mu - k\mu = \mu(1 - k) \qquad (6)$$

$$\text{error variance} = V(e) = V(y - k\,\hat{\mu}_c) = V(y) + V(k\,\hat{\mu}_c) = \sigma^2 + k^2 \frac{\sigma^2}{n} = \sigma^2 \left(1 + \frac{k^2}{n}\right) \qquad (7)$$

$$\text{PMSE} = E[(y - k\,\hat{\mu}_c)^2] = E(y^2 - 2k\,y\,\hat{\mu}_c + k^2\,\hat{\mu}_c^2) = \mu^2 + \sigma^2 - 2k\,\mu^2 + k^2\left(\mu^2 + \frac{\sigma^2}{n}\right)$$

$$= \mu^2(1-k)^2 + \sigma^2 \left(1 + \frac{k^2}{n}\right) = \text{bias}^2 + \text{error variance} \tag{8}$$

From the above equations we see that for $k = 0$ the bias of the prediction $\hat{y}$ is $\mu$ and the variance is $\sigma^2$, giving a PMSE of $\mu^2 + \sigma^2$. For $k = 1$ the bias is 0 and the variance is $\sigma^2 \left(1 + \frac{1}{n}\right)$, giving a PMSE equal to the variance. We now seek to find the value of $k$ that minimizes the PMSE. The derivative of the PMSE with respect to $k$ is given by

$$\frac{d\text{PMSE}}{dk} = 2k\left(\mu^2 + \frac{\sigma^2}{n}\right) - 2\mu^2 \tag{9}$$

From this we find that the PMSE has a minimum at

$$k = \frac{\mu^2}{\mu^2 + \frac{\sigma^2}{n}} = \frac{1}{1 + \frac{\sigma^2}{n\mu^2}} = \frac{1}{1 + \frac{1}{s^2}} \tag{10}$$

where $s$ is the SNR $s = n^{1/2}|\mu|/\sigma$. Equation (10) shows that the value of $k$ at the minimum always lies in the interval [0,1]. We see from the above derivation that there is a value of $k$ between 0 and 1 which gives a lower PMSE than either the prediction for no change ($k = 0$) or the unadjusted ensemble mean ($k = 1$). Relative to the ensemble mean, the prediction based on this optimal value of $k$ has a higher bias, but a lower variance, which is why we refer to it as a bias-variance trade-off: in the expression for PMSE we have increased the bias squared term, in return for a bigger reduction in the variance term. The PMSE of this prediction is lower than the PMSE of the prediction based on the ensemble mean because of the reduction in the term $\frac{\sigma^2 k^2}{n}$, which represents the contribution to PMSE of the estimation error of the ensemble mean. For an infinite size ensemble, this term would be zero, and the optimal value of $k$ would be 1. We can therefore see the prediction $\hat{y}$ as a small-sample correction to the ensemble mean, which compensates for the fact that the ensemble mean is partly affected by the variability across a finite ensemble.

If we could determine the optimal value of $k$ then we could, without fail, produce predictions that would have a lower PMSE than the ensemble mean. However, the expression for $k$ given above depends on two unknown quantities, $\mu^2$ and $\sigma^2$, and the best we can do is to attempt to estimate $k$ based on the information we have. The most obvious estimator is that formed by simply plugging-in the observed equivalents of $\mu^2$ and $\sigma^2$, calculated from the ensemble, which are $\hat{\mu}_c{}^2$ and $\hat{\sigma}_c{}^2$, giving the plug-in estimate for $k$:

$$\hat{k}_S = \frac{\hat{\mu}_c{}^2}{\hat{\mu}_c{}^2 + \frac{\hat{\sigma}_c{}^2}{n}} = \frac{1}{1 + \frac{1}{\hat{s}^2}} \tag{11}$$

This is the estimate of $k$ that we will use in the SPMA method. From Eq. (8) it gives predictions with a corresponding PMSE of:

$$\sigma_S{}^2 = \hat{\mu}_c{}^2\left(1 - \hat{k}_S\right)^2 + \hat{\sigma}_c{}^2\left(1 + \frac{\hat{k}_S{}^2}{n}\right) \tag{12}$$

The fact that SPMA works by introducing a bias should not be a cause for concern. Bias, in this sense, is an abstract statistical quantity. PMSE, which is minimized by SPMA, is of more relevance as a measure of accuracy.

### 3.2.1 Relation to Statistical Significance

We can relate the value of the weight $\hat{k}_S$ to the threshold for statistical significance, since statistical testing for changes in the mean of a normal distribution also uses the observed SNR, in which context it is known as the t-statistic. For a sample of size 10, two-tail significance at the 95% confidence level is achieved by signals with a SNR value of 2.262 or greater. This means that if the change in the ensemble mean gives a SNR value of greater than 2.262 then we can be 95% confident that the change in the mean is not just due to random variability caused by variability between the different ensemble members, but indicates a genuine difference between the two ensembles caused by the different forcing. From Eq. (11), a value of SNR of 2.262 corresponds to a $\hat{k}_S$ value of 0.837. All locations with $\hat{k}_S$ values greater than this are therefore statistically significant at the 95% level, while all locations with $\hat{k}_S$ values less than this are not statistically significant.

### 3.2.2 Generation of Probabilistic Predictions

Applying SPMA to a climate projection adjusts the mean. By making an assumption about the shape of the distribution of uncertainty, we can also derive a corresponding probabilistic forecast, as follows. We will assume that the distribution of uncertainty, for given values of the estimated mean and variance $\hat{\mu}_c$ and $\hat{\sigma}_c^2$, is a normal distribution. For the unadjusted ensemble mean, an appropriate predictive distribution can be derived using standard Bayesian methods, which widen and change the predictive distribution so as to take account of parameter uncertainty on the estimates of $\hat{\mu}_c$ and $\hat{\sigma}_c^2$. Bayesian methods require priors, and sometimes the choice of prior is difficult and arbitrary, but the normal distribution is one of the few statistical models that have a unique objective prior that is appropriate in the context of making predictions (see, for example, standard Bayesian textbooks such as Lee (1997) or Bernardo and Smith (1993)). This prior, often known as the Jeffreys' Independence Prior, has a number of attractive properties, including that the resulting predictions match with confidence limits. The predictions based on this prior are t distributions. If we write the probability density for a random variable $y$ that follows a t distribution with location parameter $a$, scale parameter $b$ and degrees of freedom $c$ as $St(y|a,b,c)$ then, following Bernardo and Smith, page 440 (Bernardo & Smith, 1993) this prediction can be written as

$$p(y) = St\left(y|\hat{\mu}_c, \sqrt{\frac{11}{10}}\hat{\sigma}_c, 9\right) \tag{13}$$

The location parameter (which is also the mean of the t distribution) is given by the usual estimate for the mean, $\hat{\mu}_c$, the scale parameter (which is *not* the variance of the t distribution) is given by a slightly scaled version of the square root of the usual unbiased estimate for the variance, $\hat{\sigma}_c$, and the number of degrees of freedom is given by the ensemble size minus 1. This formulation gives us probabilistic predictions based on the unadjusted ensemble mean. We then modify this formation to create probabilistic predictions based on the SPMA-adjusted ensemble mean: the distribution remains a t distribution, the location parameter is given by the SPMA-adjusted mean, the scale parameter is given in terms of the PMSE of the SPMA

prediction from Eq. (8), and the number of degrees of freedom are again given by the ensemble size minus 1. The probability density for SPMA is then given by:

$$p(y) = St\left(y|\hat{k}_S\hat{\mu}_c, \sqrt{\tfrac{11}{10}}\sigma_s, 9\right) \tag{14}$$

### 3.3 Bayesian Plug-in Model Averaging (BPMA) Methodology

The BPMA method was described and tested using simulations in a second non-peer-reviewed technical report (Jewson & Hawkins, 2009b), but again was not tested extensively on real climate data. The BPMA method is an attempt to improve on SPMA by using standard Bayesian methods to reduce the impact of parameter uncertainty on the estimate of the weight $k$. It is derived as an extension of the SPMA method as follows. Since the prediction in the SPMA method $\hat{y}$ depends on $\hat{k}_S$ and $\hat{\mu}_c$, and $\hat{k}_S$ depends on $\hat{\mu}_c$ and $\hat{\sigma}_c$, we see that the prediction $\hat{y}$ is affected by parameter estimation uncertainty on $\hat{\mu}_c$ and $\hat{\sigma}_c$.

As a result of this parameter uncertainty the reduction applied to the ensemble mean in SPMA might be too large, or not large enough, relative to the ideal reduction. Since we only have 10 ensemble members with which to estimate the reduction, this uncertainty is large. We take a standard Bayesian approach to managing this parameter uncertainty, using objective Bayesian methods, as follows. The observed values $\hat{\mu}_c$ and $\hat{\sigma}_c$ are the best single estimates for the real unknown values $\mu_c$ and $\sigma_c$, but other values of $\mu_c$ and $\sigma_c$ are also possible. Using Bayes' theorem in the usual way, we can evaluate the whole

distribution of possible values of $\mu_c$ and $\sigma_c$ by combining a prior distribution (for which we use the standard objective prior for the normal distribution, as used in Sect. 3.2.2 above) with the likelihood function for $\hat{\mu}_c$ and $\hat{\sigma}_c$ (which is derived from the 10 values). This gives a posterior probability distribution $p(\hat{\mu}_c, \hat{\sigma}_c)$, which tells us the distribution of possible values of $\hat{\mu}_c$ and $\hat{\sigma}_c$ that can be inferred from the data at that location. For each possible pair of values $\hat{\mu}_c$ and $\hat{\sigma}_c$ we can calculate an SPMA prediction $\hat{y} = \hat{y}(\hat{\mu}_c, \hat{\sigma}_c)$. We then combine the probability distribution $p(\hat{\mu}_c, \hat{\sigma}_c)$ with all possible SPMA predictions

$\hat{y}(\hat{\mu}_c, \hat{\sigma}_c)$ to calculate the expected value of $\hat{y}$, which we use as our BPMA prediction. This combination is given by the integral

$$\hat{y}_B = \iint \hat{y}(\hat{\mu}_c, \hat{\sigma}_c) \ p(\hat{\mu}_c, \hat{\sigma}_c)d\hat{\mu}_c d\hat{\sigma}_c \tag{15}$$

Relative to SPMA we are no longer using just a single prediction for $\hat{y}$ based on our best estimate values for $\hat{\mu}_c$ and $\hat{\sigma}_c$, but an average prediction based on individual predictions derived from all the possible values for $\hat{\mu}_c$ and $\hat{\sigma}_c$. This integral could

be evaluated in various different ways. We use straightforward monte-carlo integration, in which we simulate pairs of values $\hat{\mu}_c$ and $\hat{\sigma}_c$ from the distribution $p(\hat{\mu}_c, \hat{\sigma}_c)$, and calculate $\hat{y}$ for each one. We then average the many $\hat{y}$ values together to give an estimate of the expectation, $\hat{y}_B$. We tested various numbers of simulations, and found that simulating 250 pairs of values $\hat{\mu}_c$ and $\hat{\sigma}_c$ at each location was more than sufficient to give good convergence of the results. For purposes of comparison with the SPMA method we can then reverse-engineer an effective value of $k$, given by $\hat{k}_B = \hat{y}_B/\hat{\mu}_c$. The probability density of

the BPMA prediction can then be written as:

$$p(y) = St\left(y | \hat{k}_B \hat{\mu}_c, \sqrt{\tfrac{11}{10}} \sigma_B, 9\right)$$

(16)

where

$$\sigma_B{}^2 = \hat{\mu}_c{}^2\left(1 - \hat{k}_B\right)^2 + \hat{\sigma}_c{}^2\left(1 + \frac{\hat{k}_B{}^2}{n}\right)$$

(17)

### 3.4 Simulation results

Given that $\hat{k}_S$ and $\hat{k}_B$ are only estimated, there is no guarantee that the predictions from the SPMA and BPMA methods will actually have a lower PMSE than the ensemble mean, in spite of the derivation which is based on the idea of minimising PMSE. This is a common problem that arises in many statistical methods, which occurs when there is a step in the derivation in which the unknown real parameters are replaced with estimated values. To gain some insight into the possible impact of this issue we can use the standard approach of exploring the performance of SPMA and BPMA using simulations, as follows. We vary a SNR parameter from 0 to 7, in 100 steps. For each value, we simulate 1 million synthetic ensembles, each of 10 points from a normal distribution. For each ensemble we create predictions using the estimated ensemble mean, AICMA, SPMA, BPMA and statistical testing and compare the predictions with the underlying known mean, which we know in this case because these are ensembles we have generated ourselves. We calculate the PRMSE of each method relative to the PRMSE of the estimated ensemble mean. Results are shown in Fig. 3a. The horizontal line shows the performance of the unadjusted ensemble mean, which is constant with SNR, and which is determined simply by the variance of the variable being predicted and the parameter uncertainty on the ensemble mean. The red dashed line shows the performance of the SPMA method. We see that it does better than the ensemble mean for small values of SNR, up to around 1.45, and worse thereafter. For large values of the SNR its performance asymptotes to that of the ensemble mean. The worst performance is for values of SNR of around 2.5. The blue dotted line shows the performance of the BPMA method. It shows a similar pattern of behaviour to the SPMA method: it does better than the ensemble mean for small values of SNR, now up to around 1.9, and worse thereafter. For both small and large SNR values it performs worse than the SPMA method, while for a range of intermediate values it performs better. The purple dot-dashed line shows the performance of statistical testing, which gives the best predictions for the very smallest values of SNR, but the poorest predictions over a large range of intermediate SNR values. This poor predictive performance is related to the use of a high threshold that has to be crossed before any information from the ensemble is used. The green long-dashed line shows the performance of AICMA, which shows results in between statistical testing and the PMA methods. Comparing the four methods, we see there is a trade-off whereby those methods that perform best for small and large SNR values perform the least well for intermediate values. The spatial average performance on a real data-set will then depend on the range of SNR values in that dataset. Although this graph gives us insight into the performance of the various methods, and suggests that, depending on the range of actual SNR values, they may all perform better than the ensemble mean in some cases, it cannot be used as a look-up table to determine

which of the methods to use. This is because the results are shown as a function of the actual SNR value (as opposed to the estimated SNR value), and in real cases this actual value is unknown.

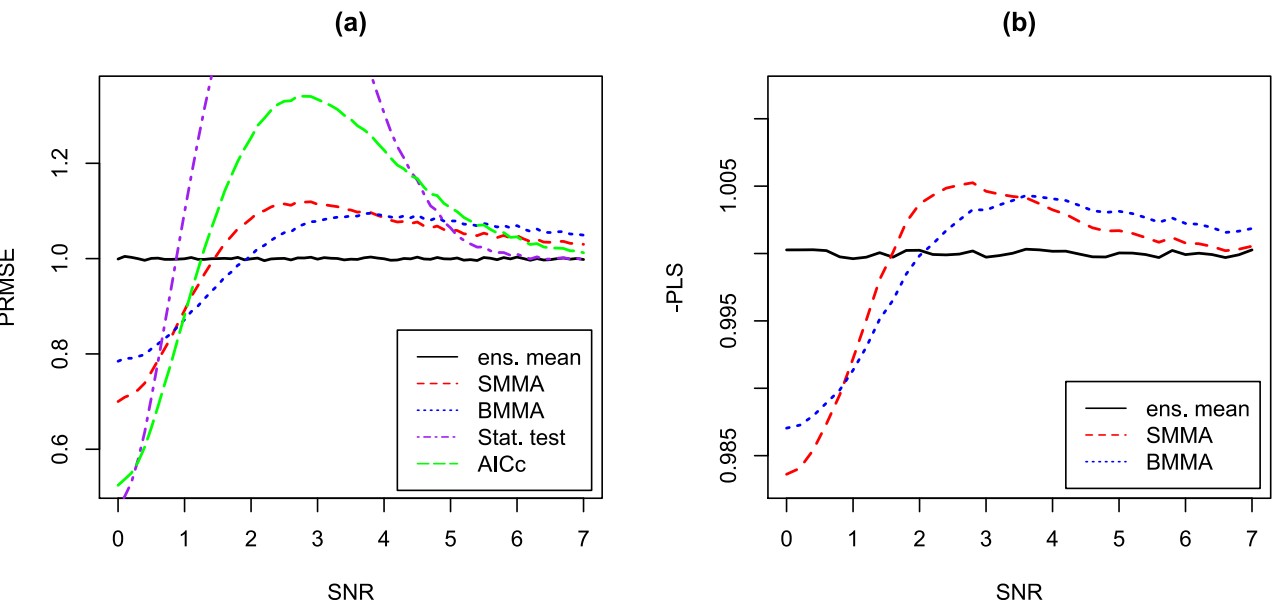

Figure 3: panel (a) shows the results of a simulation experiment for quantifying the performance of the two Plug-in Model Averaging (PMA) methods, comparing with the ensemble mean, statistical testing and AICMA. Panel (a) shows performance of point forecasts in terms of predictive root mean squared error (PRMSE). Panel (b) shows performance of probabilistic forecasts in terms of predictive log-score (PLS). The horizontal black solid line in both panels is the performance of the unadjusted ensemble mean, versus the real SNR, which would usually be unknown. The red dashed line in both panels shows the performance of the Simple PMA (SPMA) scheme and the blue dotted line in both panels shows the performance of the Bayesian PMA (BPMA) scheme. In panel (a) the purple dot-dashed line shows the performance of statistical testing and the green long-dashed line shows the performance of AICMA.

We can also use simulations to test whether SPMA and BPMA give better probabilistic predictions, for which we need to replace PRMSE with a score that evaluates probabilistic predictions. Many such scores are available: see the discussion in text-books such as Jolliffe and Stephenson (2003) and Wilks (2011). We use the score which is variously known as the log-score, the log-likelihood score, the mean log-likelihood or (after multiplying by minus one) the surprisal, or ignorance. Log-score (LS) seems to be the most widely used of these names, so we use that. Since we use the log-score in a predictive sense we call it the Predictive Log Score (PLS). PLS evaluates the ability of a prediction to give reasonable probabilities across the whole of the probability distribution. PLS is a proper score, and, according to Brocker and Smith (2007) is the only proper

local score for probability forecasts of a single variable. It also has a close relationship to measures of information in the forecast (Winkler, 1969).

Fig. 3b follows Fig. 3a, but now shows validation of probabilistic predictions using Predictive Log Score (PLS). We show the PLS values as minus one times PLS, to highlight the similarities between the results in panels (a) and (b). We only show probabilistic results for the ensemble mean, SPMA and BPMA. We see that the pattern of change in PLS from using the two PMA methods is almost identical to the pattern of change in PRMSE: for small values of SNR, the PMA methods give better probabilistic predictions than the ensemble mean, while for large values of SNR, the PMA methods give less good probabilistic predictions than the ensemble mean. The relativity between SPMA and BPMA is also the same as for PRMSE. The similarity between the results for PRMSE and PLS can be understood using the decomposition of the PLS given in Jewson et al. (2004), which shows that PLS can be written as two terms, one of which is proportional to the PRMSE.

The overall implication of these simulation results is that whether or not the FMA methods are likely to improve predictions of climate change depends on the SNR of the change. For situations in which the impact of climate change is large and unambiguous, corresponding to large SNR, such as is often the case for temperature or sea-level rise, they would likely make predictions slightly worse. However, for variables such as rainfall, where the impact of climate change is often highly uncertain, corresponding to low SNR, they may well improve the predictions.

**4 Results for RCP4.5, 2011-2040, RTOT, Winter**

We now show results for the SPMA method for the single case that was previously illustrated in Fig. 1. For this case, Fig. 4a shows values of the reduction factor $\hat{k}_S$, Fig. 4b shows the adjusted ensemble mean $\hat{k}_S\hat{\mu}_c$, Fig. 4c shows the percent change in the ensemble mean from applying SPMA, and Fig. 4d shows the absolute (unsigned) change in the ensemble mean.

In Fig. 4a we see that in the regions where the ensemble mean is statistically significant (as shown in Fig. 1d), $\hat{k}_S$ is close to 1 and the SPMA method will have little effect. In the other regions it takes a range of values, and in some regions, e.g., parts of Spain, it is close to zero. These values of $\hat{k}_S$ lead to the prediction shown in Fig. 4b. The prediction does not, overall, look much different from the unadjusted prediction shown in Fig. 1a. The changes in the prediction are more clearly illustrated by the percentage differences shown in Fig. 4c and the absolute changes in Fig. 4d. SPMA does not radically alter the patterns of climate change in the ensemble mean: it selectively identifies locations where the changes have high uncertainty and makes adjustments in those locations. The impact is therefore local rather than large-scale.

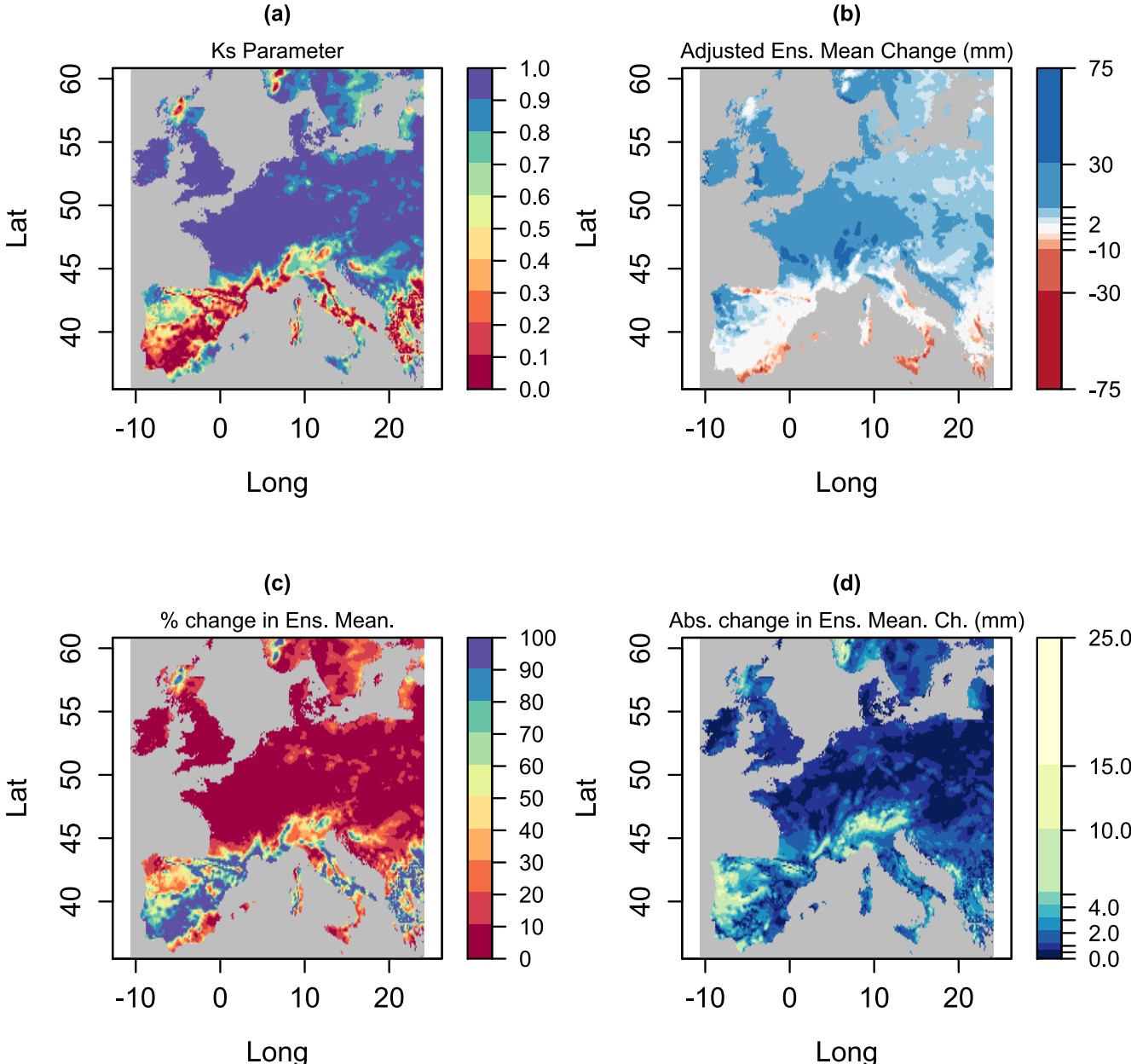

Figure 4: Various metrics derived from the EURO-CORDEX data shown in Fig. 1. Panel (a) shows the reduction parameter $\hat{k}_s$ for the SPMA method, panel (b) shows the ensemble mean reduced by the parameter $\hat{k}_s$, panel (c) shows the percent change in the ensemble mean from applying SPMA, and panel (d) shows the absolute (unsigned) change.

Figure 5a shows a histogram of the values of SNR shown on the map in Fig. 1c. There are a large number of values below 2, which correspond to non-significant changes in the ensemble mean. Figure 5b shows a histogram of the values of $\hat{k}_S$ shown on the map in Fig. 4a. Many of the $\hat{k}_S$ values are close to one, corresponding to regions where the change in the ensemble is
410 significant, and where the SPMA method will have little impact. However, there are also values all the way down to zero, corresponding to regions where the ensemble mean change is not significant, and where the SPMA method will have a larger impact.

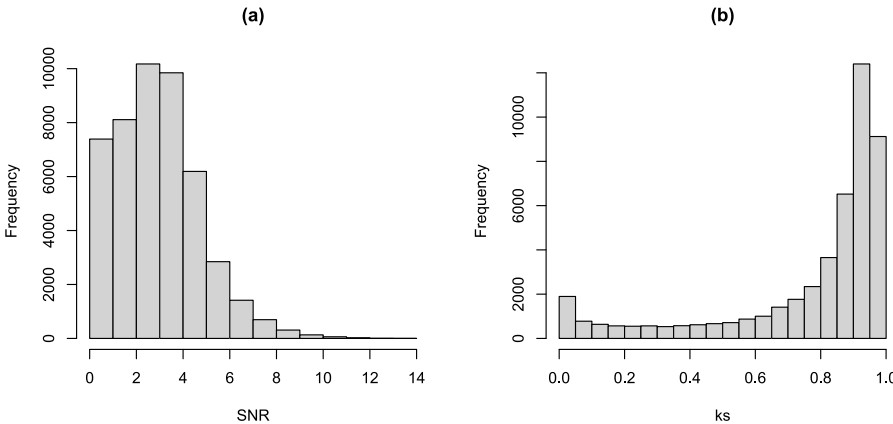

Figure 5: The left-hand panel shows the frequency distribution of the SNR values shown in Fig. 1c and the right-hand panel
shows the frequency distribution of the k values shown in Fig. 3a.

## 4.1 Cross-validation

We can test whether the adjusted ensemble means created by the PMA methods are really likely to give more accurate predictions than the unadjusted ensemble mean, as the theory and the simulations suggest they might, by using leave-one-out
cross-validation within the ensemble. Cross-validation is commonly used for evaluating methods for processing climate model output in this way (see e.g., Raisanen and Ylhaisi (2010)). It only evaluates *potential* predictive skill, however, since, as we are considering projections of future climate, we have no observations. We apply the following steps:

- At each location, for each of the 72 cases, we cycle through the ten climate models, missing out each model in turn.
- We use the nine remaining climate models to estimate the reduction factors $\hat{k}_S$ and $\hat{k}_B$.
- We make five predictions using the ensemble mean, the SPMA method, the BPMA method, statistical significance testing and AICMA.
- We compare each of the five predictions with the value from the model that was missed out.
- We calculate the PMSE over all ten models and all locations, for each of the predictions.

- We calculate the PLS over all ten models and all locations, for the ensemble mean and the PMA methods.
- We calculate the ratio of the PRMSE for the adjusted ensemble mean and statistical significance predictions to the PRMSE of the unadjusted ensemble mean prediction, so that values less than one indicate a better prediction than the unadjusted ensemble mean prediction.
- We also calculate the corresponding ratio for the PLS resultss, for the PMA methods.

For the case illustrated in Fig. 1 and Fig. 4, we find a value of the PRMSE ratio of 0.960 for the SPMA method, 0.930 for the BPMA method, 1.100 for significance testing and 0.964 for AICMA. Since the SPMA, BPMA and AICMA methods give values that are less than 1, we see that the adjusted ensemble means are, on average over the whole spatial field, giving predictions with a lower PMSE than the ensemble mean prediction. The predictions are 4%, 7% and 4% more accurate, respectively, as estimates of the unknown mean. Since statistical testing gives a value greater than one, we see that it is giving predictions with higher PMSE than the ensemble mean prediction. All these values are a combination of results from all locations across Europe. The PMSE values from the SPMA, BPMA and AICMA methods are lower than those from the ensemble mean in the spatial average but are unlikely to be lower at every location. From the simulation results shown in Sect. 3.4 above we know that the PMA and AICMA methods are likely giving better results than the unadjusted ensemble mean in regions where the SNR is low (much of Southern Europe), but less good results where the SNR is high. The final average values given above are therefore in part a reflection of the relative sizes of the regions with low and high SNR.

The values of the PLS ratio for SPMA and BPMA are 0.9983 and 0.9982, and we see that the probabilistic predictions based on the PMA-adjusted ensemble means are also improved relative to probabilistic predictions based on the unadjusted ensemble mean. The changes in PLS are small, but our experience is that small changes are typical when using PLS as a metric, as we saw in the simulation results shown in Fig. 3b.

## 5 Results for 72 Cases

We now expand our cross-validation testing from one case to all 72 cases, across four seasons, three variables, two RCPs and three time-horizons. Fig. 6 shows the spatial means of the estimates of $k$ for both PMA methods for all these cases, stratified by season, RCP, variable and time horizon. The format of Fig. 6 follows the format of Fig. 2: each panel contains 72 black circles and 72 red crosses. Each black circle is the spatial mean over all the estimates of $k$ from the SPMA method for one of the 72 cases. Each red cross is the corresponding spatial mean estimate of $k$ from the BPMA method. The horizontal lines show the means of the estimates within each sub-set. Figure 6a shows that the estimates of $k$ from both methods decrease from DJF to SON. This is because of the decreasing SNR values shown in Fig. 2a. The BPMA method gives higher $k$ estimates than the SPMA method on average, and a lower spread of values. There is no clear impact of rainfall variable on the $k$ values (Fig. 6b). Figure 6c shows higher $k$ values for RCP8.5 than RCP4.5, reflecting the SNR values shown in Fig. 2c. Figure 6d shows $k$ values increasing with time into the future, reflecting the increasing SNR values shown in Fig. 2d.

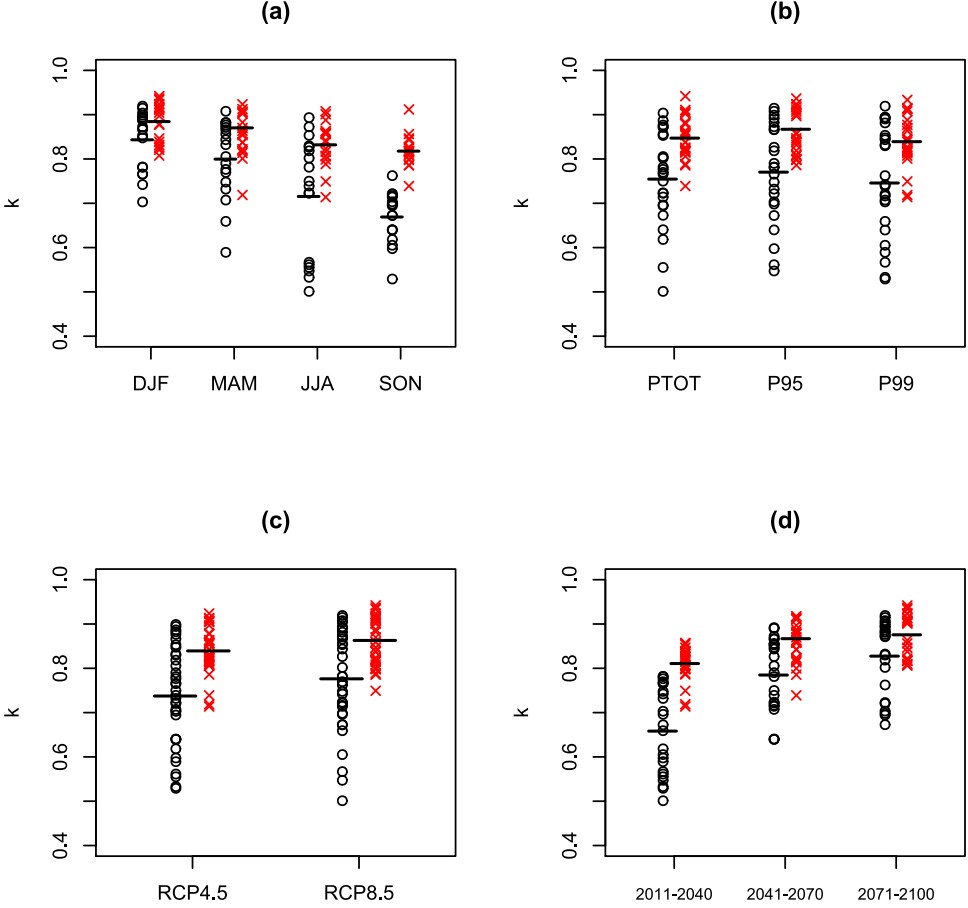

Figure 6: Each panel shows the same 72 values of the Europe-wide spatial mean of the weights $\hat{k}_S$ (black circles) and $\hat{k}_B$ (red X's) derived from the 72 EURO-CORDEX climate change projections described in the text, along with means within each subset (horizontal lines). Panel (a) shows the 72 values as a function of season, panel (b) shows them as a function of rainfall variable, panel (c) shows them as a function of RCP and panel (d) shows them as a function of time period.

Figure 7 shows corresponding spatial mean PRMSE results and includes results for significance testing (blue plus signs) and AICMA (purple triangles). For the SPMA method the PRMSE reduces (relative to the PRMSE of the unadjusted ensemble mean) for 45 out of 72 cases, while for the BPMA method the PRMSE reduces for 51 out of 72 cases. Significance testing performs much worse than the other methods, and only reduces the PRMSE for 5 out of 72 cases. AICMA reduces PRMSE for 27 out of 72 cases and so performs better than statistical testing but less well than the unadjusted ensemble mean.

Considering the relativities of the results between SPMA, BPMA, significance testing and AICMA by subset: BPMA gives the best results overall and beats SPMA for 10 out of 12 of the subsets tested. Significance testing gives the worst results and is beaten by SPMA, BPMA and AICMA in every subset. Considering the results of SPMA, BPMA significance testing and AICMA relative to the unadjusted ensemble mean by subset: SPMA beats the ensemble mean for 11 out of 12 of the subsets tested, BPMA beats the ensemble mean for 12 out of 12 of the subsets tested, significance testing never beats the ensemble mean and AICMA beats the ensemble mean for 2 out of 12 of the subsets tested. Considering the variation of PRMSE values by season (Fig. 7a) we see that the SPMA, BPMA, significance testing and AICMA all perform gradually better through the year, and best in SON, as the SNR ratio reduces (see Fig. 2a). In SON the results for SPMA and BPMA for each of the 18 cases in that season are individually better than the ensemble mean. Considering the variation of PRMSE values by rainfall variable and RCP (Fig. 7b and Fig. 7c), we see little obvious pattern. Considering the variation of PRMSE values by time period, we see that SPMA and BPMA show the largest advantage over the unadjusted ensemble mean for the earliest time period, again because of the low SNR values (Fig. 2d).

Considering results over all 72 cases we find average PRMSE ratios of 0.956 and 0.946 for the SPMA and BPMA methods respectively, corresponding to estimates of the future mean climate that are a little over 4% and 5% more accurate than the predictions made using the unadjusted ensemble mean. For significance testing we find average PRMSE ratios of 1.226, corresponding to estimates of the future mean climate that are roughly 23% less accurate than the predictions made using the unadjusted ensemble mean. For AICMA we find average PRMSE ratios of 1.02, corresponding to estimates of the future mean climate that are roughly 2% less accurate those from the unadjusted ensemble mean.

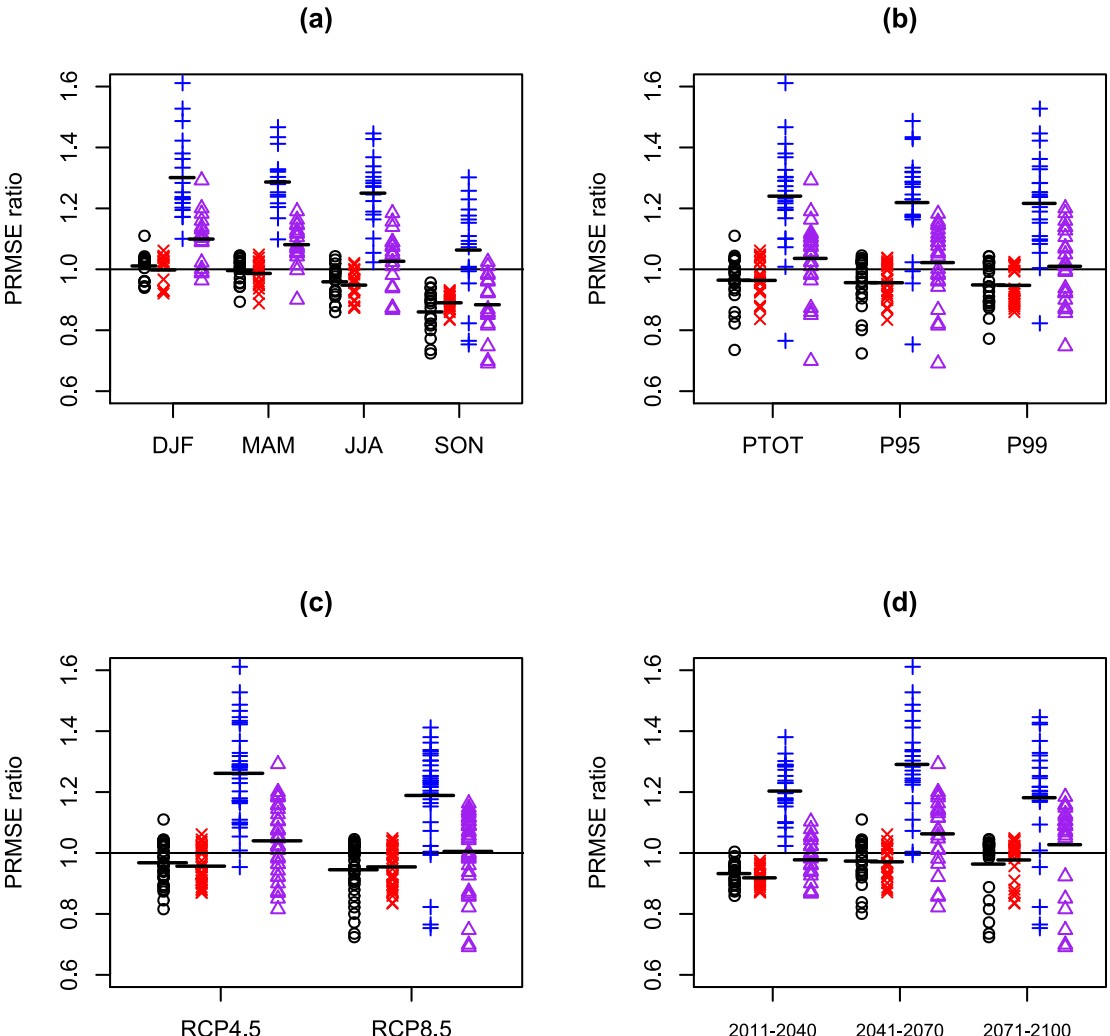


Figure 7: Each panel shows 72 values of the PRMSE ratio from the SPMA scheme (black circles), 72 values of the PRMSE ratio from the BPMA scheme (red X's), 72 values of the PRMSE ratio from significance testing (blue triangles), and 72 values of the PRMSE ratio from AICMA scheme (purple triangles), all derived from the 72 EURO-CORDEX climate change projections described in the text, along with means within each subset (horizontal lines). Panel (a) shows the 72

values as a function of season, panel (b) shows them as a function of rainfall variable, panel (c) shows them as a function of RCP and panel (d) shows them as a function of time period.

Figure 8 is equivalent to Fig. 7, but shows results for PLS i.e., evaluates the performance of probabilistic predictions. Given the poor performance of statistical testing and AICMA in terms of PRMSE we do not show their results for PLS. We see that the PLS results are very similar to the PMSE results in Fig. 7, with BPMA showing the best results, followed by SPMA, followed by the unadjusted ensemble mean. For our EURO-CORDEX data, we conclude that making the mean of the prediction more accurate also makes the probabilistic prediction more accurate, which implies that the distribution shape being used in the probabilistic predictions is appropriate.


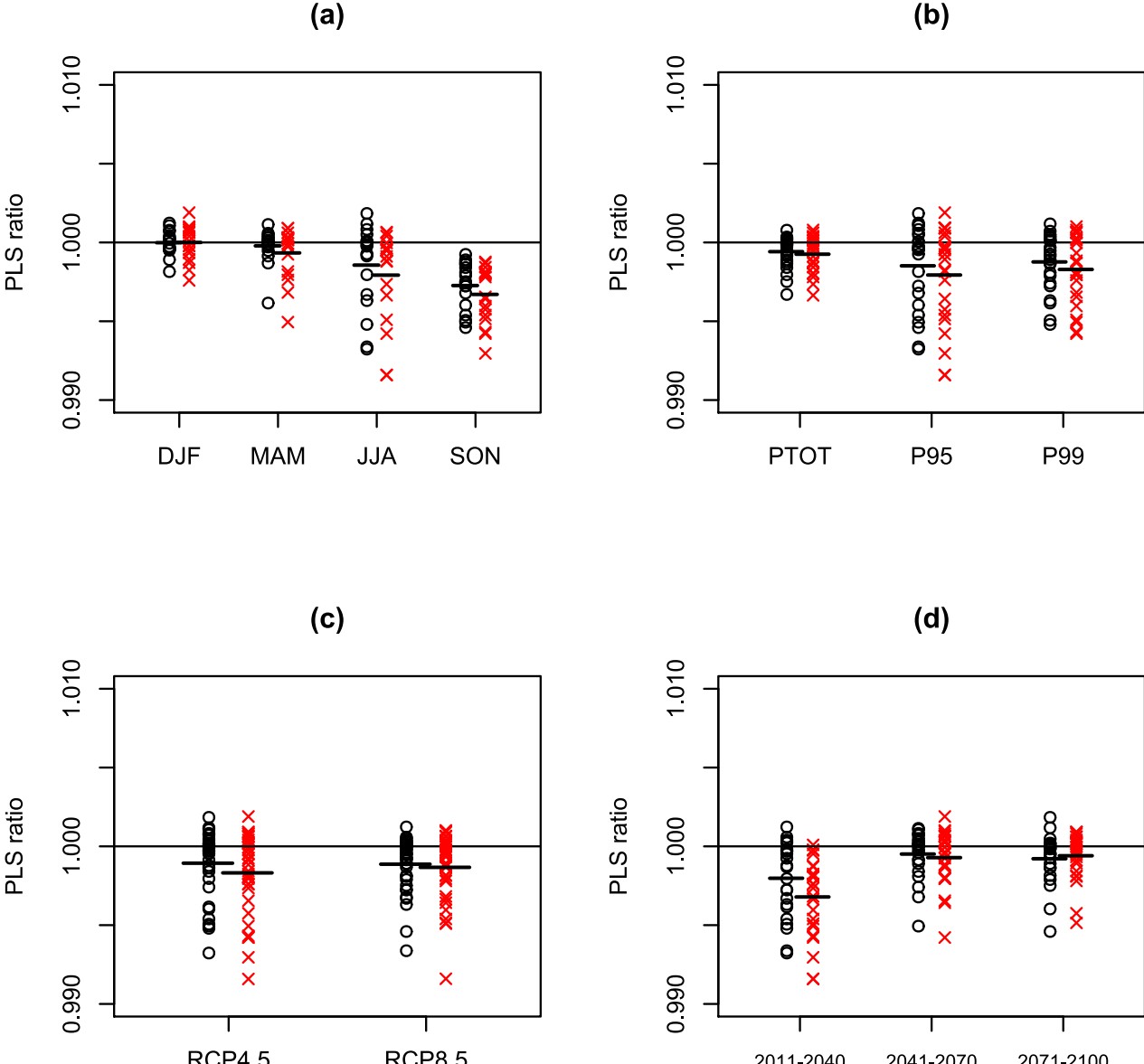


Figure 8: As Fig. 7, but now for 72 values of the predictive log score (PLS) ratio derived from probabilistic forecasts from SPMA (black circles) and BPMA (red X's).

## 5.1 Further analysis

Figure 9 shows further analysis of these results. Figure 9a shows the mean values of the estimates of $k$ for the SPMA and BPMA methods, versus the mean SNR for all 72 cases. The connection between the mean SNR and the mean $k$ is now very clear, with mean $k$ increasing with mean SNR. This panel also shows that the BPMA method gives higher $k$ values on average for all values of SNR, but particularly for low values of SNR. Figure 9b explores how much the ensemble mean is changed by the application of SPMA and BPMA, by looking at the ratio of the typical size of the ensemble mean after

adjustment to the typical size before adjustment. This metric is calculated by first squaring each prediction (for the three predictions consisting of the ensemble mean, the SPMA adjusted ensemble mean and the BPMA adjusted ensemble mean), summing the squared predictions across all locations, and taking the square root, to give the root mean square size of the predictions from each method. This gives a measure of the typical size of the predictions, for each of the three methods. The root mean square sizes for the SPMA and BPMA predictions are then compared to the root mean square size of the ensemble

mean prediction by calculating the ratio of one to the other, and Fig. 9b shows this ratio. By this measure, SPMA and BPMA give very similar results: they both apply reductions to the ensemble mean, so all the values are below one, and in both cases the impact is greatest for the cases with low SNR. These are average reductions in the size of the predictions over the whole of Europe: locally, the reduction takes values in the whole range from 0 to 1. Figure 9c shows the PRMSE, but now calculated from relative errors, relative to the spatially varying ensemble mean, which we call PRRMSE (predictive *relative*

RMSE). Values less than one for all 72 cases indicate that the PMA methods perform better than the unadjusted ensemble mean more comprehensively by this measure. The difference between these results and the straight PRMSE results arises because the locations where the PMA methods improve predictions the most on a relative basis tend to be the ones with small signals, which tend to have small prediction errors. These locations do not contribute very much to the straight PRMSE but contribute more when the errors are expressed in a relative sense. Figure 9d shows a scatter plot of the PRMSE

versus the SNR for the two methods for all 72 cases. There is a clear relation in which the PMA methods perform best for small SNR values. The relation is similar to that shown in the simulation experiment results shown in Fig. 3, but with the cross-over points (shown by vertical lines) shifted to the right, because these are now relations between averages over many cases with different underlying values for the unknown real SNR. We see that for every case in which the mean SNR is less than 2.81 the SPMA method performs better than the unadjusted ensemble mean on average, and for every case in which the

mean SNR is less than 3.02 the BPMA method performs better than the unadjusted ensemble mean on average.

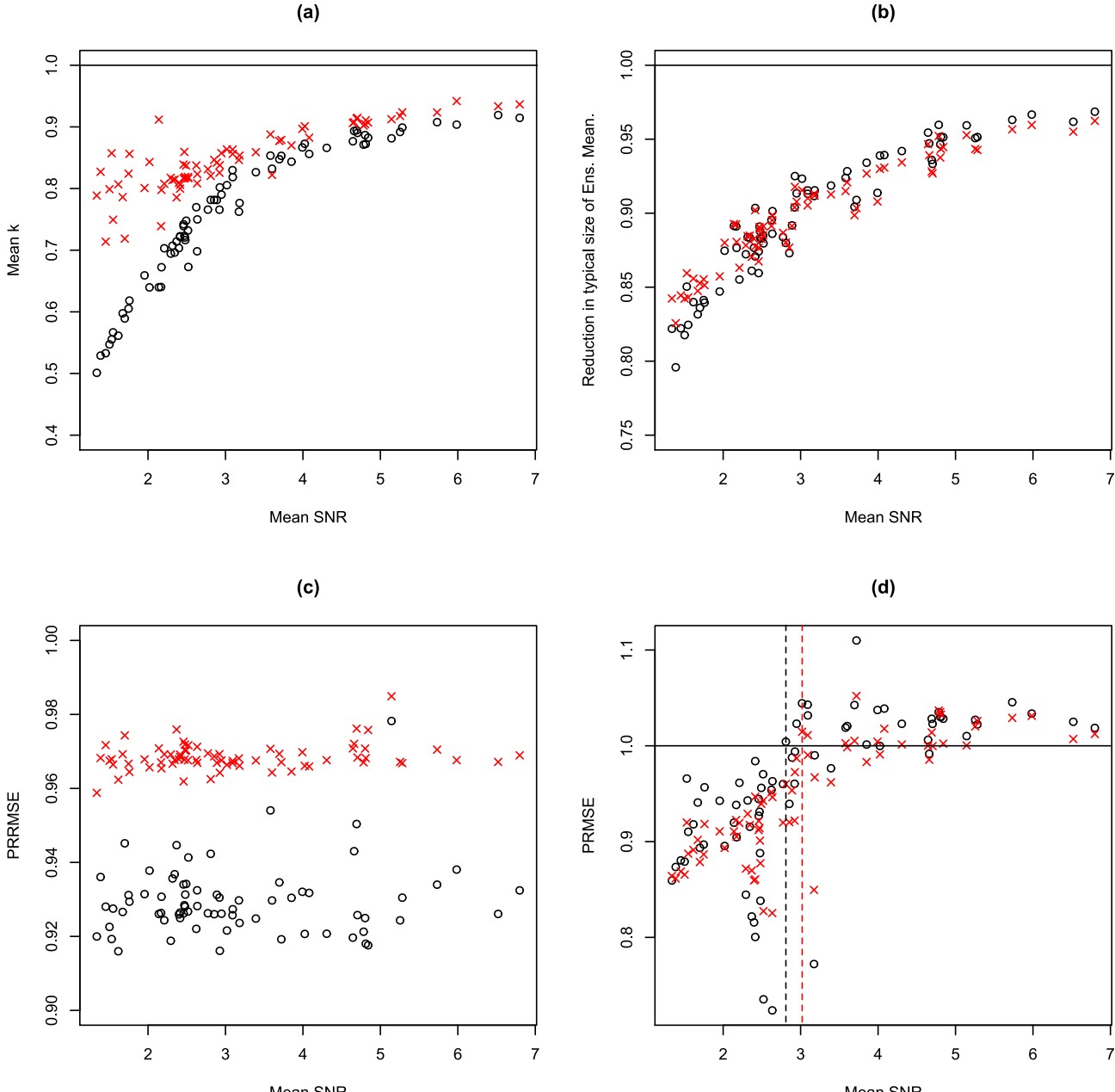

Figure 9: various diagnostics for each of the 72 EURO-CORDEX climate change projections, plotted versus mean SNR. Results from applying the SPMA scheme are shown with black circles, and results from applying the BPMA scheme are shown with red X's. Panel (a) shows mean values of the parameters $\hat{k}_S$ and $\hat{k}_B$; panel (b) shows the reduction in the typical size of the ensemble mean (calculated as described in the text); panel (c) shows the reduction in the *relative* PRMSE (the

PRRMSE) and panel (d) shows the PRMSE ratio. Panel (d) has additional vertical lines showing the cross-over points, below which the PMA results are all better than the ensemble mean results.

The results in Sect. 5 can be summarized as follows: for the EURO-CORDEX rainfall data, SPMA and BPMA give more accurate predictions on average, in both a point and a probabilistic sense, than the unadjusted ensemble mean, AICMA or

statistical testing. BPMA gives more accurate results than SPMA. The PMA methods do well because the ensemble mean is uncertain and has low SNR values at many locations. The benefits of SPMA and BPMA are greatest in the cases with the lowest SNR values.

## 6 Discussion and Conclusions

Ensemble climate projections can be used to derive probability distributions for future climate, and the ensemble mean can

be used as an estimate of the mean of the distribution. Because climate model ensembles are always finite in size, changes in the ensemble mean are always uncertain, relative to the changes in the ensemble mean that would be given by an infinite size ensemble. The ensemble mean uncertainty varies in space. In regions where the signal-to-noise ratio (SNR) of the change in the ensemble mean is high, the change in the ensemble mean gives a precise estimate of the change in the mean climate that would be estimated from the infinite ensemble. However, in regions where the SNR is low, the interpretation of the change

in the ensemble mean is a little more difficult. For instance, when the SNR is very low, the change in the ensemble mean is little more than random noise generated by variability in the members of the ensemble, and cannot be taken as a precise estimate of the change in mean climate of the infinite ensemble. In these cases, it would be unfortunate if the ensemble mean were interpreted too literally, or were used to drive adaptation decisions.

We have presented two bias-variance trade-off model averaging algorithms that adjust the change in the ensemble mean as a

function of the SNR in an attempt to improve predictive accuracy. We call the methods Plug-in Model Averaging (PMA) methods, since they use a statistical method known as plugging in. One method is very simple (simple PMA, SPMA), and the other is a more complex Bayesian extension (Bayesian PMA, BPMA). The methods can both be thought of as continuous generalisations of statistical testing, where instead of accepting or rejecting the change in the ensemble mean they apply continuous adjustment. They can also be thought of as small-sample corrections to the estimate of the ensemble mean. When

the SNR is large the ensemble mean is hardly changed by these methods, while when the SNR is small the change in the ensemble mean is reduced towards zero in an attempt to maximise the predictive skill of the resulting predictions.

We have applied the PMA methods to a large data-set of high-resolution rainfall projections from the EURO-CORDEX ensemble, for 72 different cases across four seasons, three different rainfall variables, two different RCPs and three future time periods during the 21$^{st}$ century. This data shows large variations in the SNR, which results in large variations of the

extent to which the ensemble mean is adjusted by the methods.

We have used cross-validation within the ensemble to test whether the adjusted ensemble means achieve greater potential predictive skill for point predictions and probabilistic predictions. To assess point predictions we used predictive mean

squared error (PMSE) and to assess probabilistic predictions we used predictive log-score (PLS), which are both standard measures. For both measures, we compared against results based on the unadjusted ensemble mean. For PMSE we have additionally compared against results based on statistical testing and small-sample Akaike Information Criterion model averaging (AICMA, a standard method for model averaging). We emphasize that these calculations can only tell us about the potential accuracy of the method, not the actual accuracy, since we cannot compare projections of future climate with observations. On average over all 72 cases and all locations, the PMA methods reduce the PMSE, corresponding to what is roughly a 5% increase in potential accuracy in the estimate of the future mean climate. For the SPMA method, the PMSE reduces for 45 of the 72 cases, while for the BPMA method the PMSE reduces for 51 out of 72 cases. Which cases show a reduction in PMSE and which not depends strongly on the mean SNR within each case, in the sense that the PMA methods perform better when the SNR is low. For instance, the winter SNRs are high, and the average PMSE benefits of the PMA methods are marginal. The autumn SNRs are much lower, and the PMA methods beat the unadjusted ensemble mean in every case. Significance testing, by comparison, gives much worse PMSE values than the unadjusted ensemble mean, and AICMA gives slightly worse PMSE values than the unadjusted ensemble mean. Considering probabilistic predictions, the PLS results also show that the PMA methods beat the unadjusted ensemble mean.

The ensemble mean can be used as a standalone indication of the possible change in climate, or as the mean of a distribution of possible changes in a probabilistic analysis. We conclude that in both cases, when the ensemble mean is highly uncertain, the PMA-adjusted ensemble means described above can be used in its place. Applying PMA has various advantages: (a) it reduces the possibility of over-interpreting changes in the ensemble mean that are very uncertain, while not affecting more certain changes, (b) relative to significance testing, it avoids jumps in the ensemble mean change, and (c) when the SNR is low, it will likely produce more accurate predictions than predictions based on either the unadjusted ensemble mean or statistical testing. In addition to the above advantages, relative to statistical testing the PMA-adjusted ensemble mean reduces the likelihood of false negatives (i.e., not modelling a change that is real) and increases the likelihood of false positives (i.e., modelling a change that is not real but is just noise). Whether this is an advantage or not depends on the application, but is beneficial for risk modelling. This is because the goal in risk modelling is to identify all possible futures, and hence no changes should be ignored if there is some evidence for them, even if those changes are not statistically significant.

**Author Contribution**

SJ designed the study and the algorithms, wrote and ran the analysis code, produced the graphics and wrote the text. GB, PM and JM extracted the EURO-CORDEX data. MS wrote the code to read the EURO-CORDEX data. All the authors contributed to proof-reading.

**Acknowledgements**

The authors would like to thank Tim DelSole, Ed Hawkins, Arno Hilberts, Luke Jackson, Shree Khare, Ludovico Nicotina and Jeremy Penzer for interesting discussions on this topic, Francesco Repola from CMCC for assisting with data extraction, and Casper Christophersen, Marie Scholer and Luisa Mazzotta from EIOPA for arranging the collaboration with CMCC. We

also thank EURO-CORDEX, and the climate modelling groups (listed in Table 1 of this paper) for producing and making their model output freely available.

## Data Availability

The EURO-CORDEX data used in this study is freely available. Details are given at https://euro-cordex.net.

**Competing Interests**

MS works for RMS Ltd, a company that quantifies the impacts of weather, climate variability and climate change.

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
