# Peer review of "Improving the Potential Accuracy and Usability of EURO-CORDEX Estimates of Future Rainfall Climate using Mean Squared Error Model Averaging"

_Nonlinear Processes in Geophysics, 2021_

## Author Comment (AC3)

Improving the Potential Accuracy and Usability of EURO-CORDEX Estimates of Future Rainfall Climate using Frequentist Model Averaging

Stephen Jewson, Giuliana Barbato, Paola Mercogliano, Jaroslav Mysiak, and Maximiliano Sassi

Response to Reviewer 1

Dear Reviewer 1,

Many thanks for taking the time to read our paper and make helpful comments, which have definitely improved it. I have pasted your comments below, greyed them out, and responded to them in black. Please note that I have made some additional changes based on reviewer 2's comments, and based on some additional literature on the topic of model averaging that I have found recently (in statistics and economics journals). In particular, as a result of what I've found in the statistics and economics literature, I've changed the names of the methods so that MMA becomes PMA, which means that SMMA becomes SPMA, and BMMA becomes BPMA.

The paper presents two methods to adjust ensemble mean of variables projected by climate models (CM) and compares their performances against two other adjusting approaches (i.e., conventional Akaike model averaging and statistical testing) and unadjusted mean, considering change along different future time frames, seasons, precipitation variables and RCP scenarios over the whole Europe.

The two proposed methods (MMA) have a common derivation based on minimisation of the predictive mean squared error.

The paper discusses the relative advantages of all the considered methods and shows that the application of MMA is particularly advantageous when the uncertainly of a given change is high due to small predicted changes and large spread among the CM signals (in such cases rejection of a null hypothesis of no change is usually the outcome of statistical tests).

As a general comment it is my opinion that the paper is timely, and results are of interest for NPG readers. However, the readability of the paper is not fluent and can be improved by a careful proofreading, since there are many parts of the manuscript that I needed to read and read again to understand the underlying message.

I have taken this comment to heart, and rewritten various sections. In particular:

a) the introduction

b) the section on probabilistic prediction (3.2.2)

c) the section on Bayesian prediction (3.3)

d) I've added a section on probabilistic skill scores (3.4)

Apart from these aspects I have only some minor issues, that are listed below.

The two presented methods (MMA) have been previously published in technical reports by the first author ((Jewson & Hawkins, 2009a, b; see reference list in the manuscript), as credited in Section 3.5. I suggest anticipating this information by providing proper credits in previous sections (e.g. sections 3.2 and 3.3) and to remove Section 3.5.

Thanks for this suggestion. I have moved these sections to 3.2 and 3.3, which I agree is the more logical place for them.

Line 243. "the scale parameter" shoud be "the square of the scale parameter", "numbers" is "number"

Thanks for noticing those mistakes. I have corrected both.

Lines 251-256. Clarify if the objective prior is adjusted or unadjusted. Moreover the implementation of the Bayesian approach should be better explained.

The objective prior is the standard objective prior for this distribution, with no adjustment. I have significantly lengthened section 3.3, to explain the Bayesian approach in more detail, added some equations to make it clearer, and added another citation.

Line 263. Please expain the rationale of statistical testing and conventional AICc model averaging.

I have now explained the interpretation of the statistical testing in section 3.2.1, and the first paragraph in section 3 now explains AIC model averaging in more detail, and gives citations to two text-books that cover the topic.

Line 295. Similarly to previous comment: Please explain the rationale of Predictive mean log-likelood

Based on this comment, and comments from reviewer 2, I have now added a paragraph in section 3.4 which explains the choice of score for evaluating the probabilistic forecasts. The score we are using is very commonly used, but has many different names in the literature (unfortunately). After reading around I've concluded that the name 'log-score' is more commonly used than 'mean log-likelihood', and so have changed the name from PMLL to predictive log-score (PLS) throughout.

Line 327. "Fig. 4d …." I do not unsderstant what is plotted in such subplot.

I have over-hauled figures 1 and 4, and I think they are much easier to understand now. Fig 4d now shows the absolute (unsigned) change in the ensemble mean change, in mm. So it shows the impact of applying the SMMA/SPMA method. The impact, when measured in mm, is largest in places where there is both a large rainfall change predicted by the ensemble mean, and that change is fairly uncertain.

Fig 4. Put unit of measures as in Fig. 2

Thanks. I have added the units now.

Line 367. "8%" would be "7%"

Thanks.

Line 443. "…. Root mean squared size …" please explain better

I have rewritten this section to explain how this is calculated.

Line 454. "Fig. 2" should be "Fig. 3"

Thanks for noticing that.

Fig 9c. Adjust the y-axis label

Thanks for noticing that.

Best regards,

Steve Jewson

---

## Author Comment (AC5)

Improving the Potential Accuracy and Usability of EURO-CORDEX Estimates of Future Rainfall Climate using Frequentist Model Averaging

Stephen Jewson, Giuliana Barbato, Paola Mercogliano, Jaroslav Mysiak, and Maximiliano Sassi

Response to reviewer 2

Dear Reviewer 2,

Many thanks for taking the time to read our paper in such detail and make helpful comments. I very much appreciate the time and effort it must have taken. I have pasted your comments below, greyed out, and then responded to them in black.

This manuscript describes two variants of a correction method for the ensemble mean that aims to minimize predictive root mean square error (PRMSE). The methods are then applied to three rainfall variables in the EURO-CORDEX regional climate ensemble.

The correction method consists in adding a scaling factor k to the estimator of the ensemble mean, and minimizing the PRMSE with respect to k. Note that this occurs at the expense of the expected bias, and hence is referred to as a bias-variance trade-off. One of the goals is to provide an alternative for statistical significance testing for a nonzero signal, as the latter gives rise to spatial discontinuities.

General comments:

The manuscript addresses a relevant question and is well-structured. The examples clearly show in which situations the method reduces the PRMSE. Some sentences are a bit long or hard to parse but otherwise it is well-written.

I have tried to make it easier to read. In particular I have rewritten:

a) the introduction

b) the section on probabilistic prediction (3.2.2)

c) the section on Bayesian prediction (3.3)

and I've added a section on probabilistic skill scores (3.4). Plus lots of other small edits.

The chosen scaling approach seems quite ad-hoc to me, and this could be improved by a better motivation and/or more context. If I understand it well, the work takes an operational correction method that seems to work well in practice, and tries to motivate it in a more scientific way and apply it in the context of climate change.

Secondly, I do not find the term "Model Averaging" appropriate here. In the broadest sense of the word it can indeed be seen as model averaging (with equal weights) and a fictitious "zero change" model. However the terms "calibration" or "post-processing" are more appropriate for this kind of approach.

Let me answer these two together (and sorry it's such a long answer). I've spent some time digging further into the literature on this topic of how to adjust predictions to make them more accurate. There has been lots of work in this area by economists and statisticians. Much of it has been published more recently than the Burnham and Anderson 2002 textbook that I was relying on. Examples of recent papers on these topics in economics and statistics (and this is just a small subset) would include:

1) Hjort, N.; Claeskens, G. (2003). "Frequentist model average estimators". *Journal of the American Statistical Association*. **98**: 879–899.
2) Hansen, B. (2007). "Least Squares Model Averaging". *Econometrica*. **75**: 1175–1189.
3) Hansen, B.; Racine, J. (2012). "Jackknife Model Averaging". *Journal of Econometrics*. **167**: 38–46.
4) Liu, C. (2014). "Distribution theory of the least squares averaging estimator". *Journal of Econometrics*. **186** (1): 142–159.
5) Charkhi, A.; Claeskens, G.; Hansen, B. (2016). "Minimum mean squared error model averaging in likelihood models". *Statistica Sinica*. **26**: 809–840.

There have also been two more textbooks:

1. Claeskens, G; Hjort, N (2008). *Model Selection and Model Averaging*. CUP
2. Fletcher, D *(2019). Model Averaging. Springer.*

What I've learnt from reading these papers and books is:

- In statistics and economics, the methods that I'm using fall into a general class of methods known as 'frequentist model averaging' (FMA). I have now mentioned that in the text and used it in the title.
- FMA methods that focus on MSE are known either as 'minimum mean squared error model averaging' (which is similar to what I called them originally) or 'least squares model averaging'.
- The specific method that I call MMA is discussed in many of the papers on this topic, but doesn't have a consistent name or abbreviation. It is generally referred as the plug-in estimator. The abbreviation MMA is used already in the field of frequentist model averaging, and refers to 'Mallows model averaging', which is something different. Given that, I've changed the name I use from MMA to PMA (plug-in model averaging).
- The method I call BMMA seems to be new. I've changed BMMA to BPMA (Bayesian plug-in model averaging) to be consistent.
- The method I call AICc model averaging is considered a standard method in the literature, and so remains a good basis for comparison. I've called it AICMA.
- Responding directly to the reviewer's comments:
  - the idea of weighting two estimators or predictors together in some way seems to be a standard idea in statistics. I've tried to motivate it a little better by referring to more of the literature listed above. In a sense all model averaging is a bit ad-hoc, and is justified only by whether it works or not.
  - wrt whether to call the methods model averaging…certainly in statistics and economics these kinds of methods seem to be called model averaging. To be absolutely sure, I emailed the econometrician Bruce Hansen (author of 3 of the papers listed above), explained to him what I am doing, and asked him whether he thought calling it model averaging was appropriate. He said he thought it was appropriate. Using the same terminology as used in statistics and economics will hopefully be useful for readers who want to follow up on that literature. For instance, googling 'frequentist model averaging' leads to lots of useful and relevant results, as does 'least squares model averaging' and 'plug-in model averaging'. So I have kept the name model averaging.
  - I'm afraid I don't understand the comment about equal weights. The weights at each location are data dependent and depend on the signal-to-noise ratio, and that's a critical part of the method. Perhaps this comment is by analogy to Bayesian model averaging, in which there would be an additional prior weight. As this is a frequentist method, there are no prior weights (or one could say that the prior weights are equal).

The method is proposed as an alternative to significance testing, which introduces spatial discontinuities. This seems like an unfair comparison though, as significance testing provides maps of significant climate change as a qualitative indicator (e.g. shading), but the resulting discontinuous fields are not typically used in impact models; or else please provide references.

I entirely agree. My intention was not to present FMA methods (using the new terminology) as direct alternatives to significance testing, or vice versa. Significance testing does something useful, and FMA does something useful, it's just that they do different things. They are both ways to 'manage' uncertainty, in different situations. I'm just trying to use significance testing as a reference point to explain the issues, and explain the need for FMA, since significance testing is generally well understood by the community, and in my experience tends to be the first thing that people think of when the question of uncertainty comes up. For instance, the question: "but is it statistically significant?" comes up very frequently, even though for impact models one can argue that it's not the most relevant question: what is perhaps more relevant is the best estimate you can make, even if the signal is not statistically significant, which is where model averaging comes in.

I have tried to rewrite the text to make this clearer. I still include quite a bit of discussion about statistical testing, since the precise reasons why statistical testing is not a good method for generating results that can be used in impact models provide good justification and explanation for FMA. Also, I think it would be odd if I didn't discuss significance testing, and the differences between significance testing and FMA, given how similar they are.

In general the manuscript would benefit from a better motivation in the form of a concrete example or impact model for which the reduced PRMSE from this method provides a tangible benefit over using the uncorrected ensemble mean. Perhaps an example from agriculture, urban hydrology, infrastructure planning… ?

Following the suggestion, I have added some discussion in the introduction about how the results could also be used for the applications suggested by the reviewer.

The general idea and motivation for this method is that it is an easy way to make uncertain climate projections more accurate. If climate projections are useful, then presumably more accurate climate projections are even more useful: it seems to me that this is more or less an axiom of climate research and hopefully doesn't need too much more motivation. The measures of accuracy that we use, PRMSE and PMLL/PLS (more about PMLL/PLS below) are standard in climate research. We have to deal with potential accuracy, because we have no observations for the future, but that's also standard. So we have used standard methods, and we have succeeded in demonstrating that we can make uncertain climate projections more accurate.

These methods are a much simpler and cheaper way to increase accuracy of climate projections than, for example, running larger ensembles or higher resolution climate models, or developing better climate model parametrisations (even though all of that should be done too, although that takes years, not seconds).

Specific comments and questions:

If you want to use the current title please provide strong arguments that the potential accuracy and usability is improved. This depends on your definition of accuracy (since the bias is increased and this might be detrimental for many use cases). Do you use a proper score? How does it affect other known scores such as the Brier score?

I agree that these are bold statements, and should be well justified. I would hope that the results we present demonstrate clearly that the potential accuracy is improved, for both point and probabilistic predictions. The scores we use, PRMSE and PMLL/PLS, are both very standard scores. In statistics,

they seem to be the most standard scores for point forecasts and probabilistic forecasts, respectively. One possible point of confusion is that MLL goes by many other names, such as 'log score', 'ignorance' (after multiplying by minus 1), or just 'log-likelihood'. We have now discussed probabilistic scoring in the paper in detail and changed the name from PMLL to predictive log-score (PLS), which seems a bit more standard. PMLL (or PLS) is indeed a proper score, and is discussed in the many references and textbooks on the scoring of probabilistic forecasts, such:

Good, I. J., 1952: Rational decisions. *J. Roy. Stat. Soc.*, **XIV ,** 107–114.

Brocker, J and Smith, S, 2007: Scoring Probabilistic Forecasts: The Importance of Being Proper, Weather and Forecasting, volume 22.

In the latter reference, it is the first score discussed, and in the conclusion the authors (who call it 'ignorance' rather than PMLL or PLS) conclude that it "*is effectively the only proper local score for continuous variables".*

One can say that bias is detrimental, and one can also say that variance is detrimental. Ideally both would be small, but unfortunately there is a trade-off between them. Because of that optimising for one of the two is generally not a good idea, because it can lead to large forecast errors. The usual way to manage this trade-off is via RMSE as a single score that combines the two, and that's what underlies our approach.

The argument we present for usability being improved is that projections that suffer from a lot of randomness are difficult to use, because they vary from location to location, and are more liable to change from one set of projections to the next. This undermines their credibility and might also lead to adaptation decisions based on this randomness. I have extended the discussion to explain more clearly why we believe that more accurate forecasts with less randomness are more usable.

L34: Aren't the correlations, present in ensembles, very relevant for catastrophe models? Doesn't one lose much of this information when moving to probabilities?

Yes, they are very relevant. The topic of exactly what information to take from climate models and how to extract it, for different types of impact models, is a very complex one, and not fully explored or understood at this point I would say. There are pros and cons to using events directly vs converting events into statistical representations. My impression is that both approaches are useful, and which to use depends very much on the application. With respect to correlations, one can say a few things:

- Catastrophe models themselves simulate weather and climate correlations. This is certainly an important component of the models.
- The change in the ensemble mean contains correlations, and if the ensemble mean is being used in a catastrophe model, those correlations will feed into the catastrophe model
- If the ensemble mean is highly uncertain, then some of those correlations will be spurious (i.e. noise). The FMA-adjusted ensemble mean will tend to eliminate some of these spurious correlations.
- When using a probability representation of an ensemble, the correlations between anomalies of different members in the climate model ensemble can be represented in a correlation matrix and fed into the catastrophe model. So no correlations need to be lost in a probabilistic approach.

L98: You perform cross-validation in the ensemble, and elsewhere in the manuscript you assume that any model correlations or biases should be tackled first, before applying the methods. Did you do this for the ensemble and if not, what are the implications for cross-validation?

With respect to model-to-model correlations: we do not account for these. There has been lots of work in the literature on accounting for model correlations (we now include 9 citations to work in that area), but it remains a difficult and controversial topic and no methods have become established as standard, or generally accepted as being highly reliable. Several authors have expressed doubts as to whether attempting to correct for correlations is really beneficial. In my experience, many applications of climate model data do not account for model correlations at this point. Given the state of the science right now, I think it's reasonable for us not to try and account for model correlations in this study. In terms of implications for our results: I don't think it would make much difference to our results if we did account for correlations in some way. The SPMA and BPMA methods work for the very simple reason that there are a large number of locations with low signal to noise ratio. Signal and noise would both change if we applied correlations, but in different ways in different locations, and I see no reason why they would change so fundamentally that there would no longer be a large number of locations with low signal to noise.

We use differences between different runs of the climate model, rather than absolute values. If there is a consistent bias in the model, then that disappears from the difference because it cancels. This is a simple way to account for biases. More complex methods are of course possible, but, similarly to the question of accounting for model correlations, this is a bit of a controversial topic. There are many different methods that have been tested, none has emerged as a standard, and it seems unresolved at this point as to whether they help. Again I don't think it would make much difference to the conclusions, for the same reasons as above.

L116: Did you use the historical experiment for the 1981-2010 baseline? This period includes a few years of the RCP scenarios (the 2005-2010 period), did you always use the corresponding RCP data in the comparison or did you choose the same baseline for both RCPs?

We used the corresponding RCP data.

Fig. 1: the image quality of the maps a-c is bad (compression artefacts) and I cannot easily distinguish positive from negative change. It would be useful if 0 change has a clearly identifiable color (e.g. white).

Yes. I've now improved my plotting skills and have completely over-hauled figures 1 and 4. They should be much easier to read now. In 1a and 4b zero is now given by white.

L169: Here you mention model dependence should be addressed. You could perhaps mention briefly how this could be done (cluster analysis, expert knowledge on the models, ...?) and whether you do it for your example. limate mode

I'm not sure if we are using terminology in the same way, but for me this is the same question as the L98 question above (when you said "model correlations" above I assumed that meant "model dependence"). I have added several more citations to papers dealing with model dependencies, and have rewritten this section to make it clearer. I haven't included any additional review of different methods for model dependencies, just because that didn't seem appropriate in a paper that's not about model dependencies.

L173: "the distribution they are sampled from perfectly accounts for their biases". The meaning of this sentence is unclear to me. Do you mean the model errors are on average unbiased? Or the model biases compensate for each other (the biases of the models, aggregated over the ensemble, behave like a statistical error and not a systematic one)?

I mean that the ensemble captures the real distribution of uncertainty i.e, the ensemble is neither biased, nor too narrow or too wide. I have rewritten those sentences to clarify that.

L180: Any reason why you don't use the unbiased estimator (divide by n-1)?

I do actually use the unbiased estimator (see the statement on line 203). The expression on line 206 is the expression for the standard error, which always uses 1/n.

(see, e.g., https://en.wikipedia.org/wiki/Standard_error).

L200: Could be unclear whether the square is inside the E(...) function, maybe add brackets?

Yes, good point, thankyou. I've added brackets as suggested.

L230-245: This might be clarified by adding an equation for the distribution here.

Yes, good idea. I've done that and I think it does clarify it a lot. I've also added the corresponding equations for BPMA in section 3.3.

L246-256: Section 3.3: seems a bit brief to me, again I feel like some equations could clarify this. Do you approximate a marginal distribution (compute an integrate) by sampling. please make this explicit. Why 250 values?

I've written this section quite substantially to clarify it. I've added 3 equations, one for the integral you mention, and two that define the resulting probabilistic prediction. We use 250 values to get good convergence…I've added that comment too.

L258-259: I find this sentence a bit strange, didn't you just derive these estimates to minimize PRMSE? Here you just verify your method with a numerical experiment.

Yes, this is a subtle point. It's something that runs through the whole of frequentist statistics, though, rather than being particular to this derivation. Equations are often derived using the real unknown values, but at the last minute you have to substitute in estimates because that's all you've got. When you do that, it's not necessarily the case that the equation still holds (and it never works as well as you would have hoped) and to be thorough you have to do simulations to check how well it really works. Ideally you then derive a correction that accounts for this problem, but that's incredibly hard, and I don't think anyone's done it in this case. I've been trying, on and off, for about 15 years, and failed, but in the end the BPMA approach resolves this problem, to some extent, in a different way. The various references to FMA given above all use simulations, in the same way that I do, to understand the real behaviour of the methods they discuss.

The most common example of this issue I can think of is the well-known expression for standard error, which is sigma-over-root-n. The sigma should be the real sigma for this equation to be true. But we don't know the real value of sigma so in practice people always put in the estimate for sigma without a thought (as I do on line 206…that's what the equality becomes an approximate equality).

L282: how does it behave as a function of the estimated SNR value?

Yes, very good question. That's what we'd love to know. But I don't think there's a way to figure that out. Every attempt at figuring that out goes round in circles. Part of the problem is that the estimated SNR value corresponds to many real SNR values, each of which has a different performance. If we knew the performance as a function of the estimated SNR value then I think we could create a method that would *always* beat the ensemble mean, and that would have replaced the ensemble mean long ago.

L309: Section 3.5: Move this to earlier in the manuscript

Done

Fig. 4 is very confusing to me, or there was something I didn't understand. Am I correct in understanding that the precipitation reduction is reduced quite a lot (so the drying is less severe) in the southeast of Spain? But it seems like the SNR was quite high there, so why this large change? I think it's unclear partially due to the choice of the color scale. I find it hard to read the SNR map in Fig. 1; the shades of orange/pink are too similar.

Yes, I agree that this graph is confusing. I have changed panels 4c and 4d to make it simpler to understand.

4c now shows the % change (in the change in rainfall) from applying the method. It all makes sense: the % changes are largest in southern europe, which corresponds to the lower S/N, the lack of significance and the small values of the k parameter. However, there are some exceptions, such as SE Spain, as you note. In SE Spain the s/n is higher, and the changes are borderline significant. This leads to a much lower % change in the ensemble mean in 4c.

4d now shows the absolute (unsigned) change from applying the method. The method always moves changes towards zero. The size of these absolute changes depends on both the % and the actual change. In SE Spain, the absolute changes are quite small. The adjusted prediction give slightly less of a reduction in rainfall in that region (i.e, slightly more rain than the ensemble mean gives, but still less than the baseline).

L377: Small changes when PMLL is used as a metric: is this also found in literature?

I've haven't been able to find any reference to that, but we also see that in the simulation results in Fig 3d.

L431 SSMA -> SMMA

Thanks.

Fig 9: Figure labels are not present (a-d)

Thanks, fixed.

The y labels don't match the description and this is confusing. What is the "signal" in panel b? Mention what PRMSRE stands for in panel c.

In the text, I've rewritten the explanation of how this figure was calculated. I've also changed the y axis label and the caption. Hopefully that makes it clearer.

L444: Root mean square size: please clarify, why are the values in the figure smaller than 1?

This has now been explained more clearly in the text. The reason it is less than 1 is that it shows the ratio of the typical sizes of the ensemble mean before and after adjustment (with after divided by before).

L514: between scenarios? I guess it's normal that there can be a jump, did you mean something else (between time periods)?

Yes, I meant different scenarios in time i.e., projections for 2050 and then 2060. But 'scenarios' is not a good word for that. I've deleted the phrase 'in space and between scenarios'.

L518: "Falsely identifying a change as being due to climate change": this is related to (erroneous) attribution, which is something else than a false positive (detecting change where there is none expected).

I'm not sure I 100% understand this point, but I have rewritten the sentence so that it doesn't mention detecting changes, but just talks about modelling changes. I'm not trying to make any point about formal detection.

L519: Please show how this is beneficial for risk modelling or add a reference.

I have now explained why this is beneficial for risk modelling: it's because risk modelling is all about identifying all possible futures, weighted by the level of evidence. This means still considering possible changes even if they are not statistically significant.

Best regards,

Steve Jewson

L518: "Falsely identifying a change as being due to climate change": this is related to (erroneous) attribution, which is something else than a false positive (detecting change where there is none expected).

---

## Author Comment (AC6)

Improving the Potential Accuracy and Usability of EURO-CORDEX Estimates of Future Rainfall Climate using Frequentist Model Averaging

Stephen Jewson, Giuliana Barbato, Paola Mercogliano, Jaroslav Mysiak, and Maximiliano Sassi

Response to community comment 1,

Dear Rasmus,

Many thanks for taking the time to read our paper and make helpful comments. I have pasted your comments below, greyed out, and then responded to them in black.

Discussions about *estimation uncertainty* is really timely and important, and I was pleased to come across this paper. One question I have is if this is the same concept that my group has tried to cope with that we call "*the law of small numbers*". We use large multi-model ensembles and downscale them with empirical-statistical donwscaling since we find that RCM-based ensembles tend to be too small, especially since they are not independent and involve many of the same GCM simulations. This is demonstrated in Mezghani A., A. Dobler, R. Benestad, J.E. Haugen, and K.M. Parding (2019), Sub-sampling impact on the climate change signal over Poland based on simulations from statistical and dynamical downscaling, J. Appl. Meteor. Climatol., 0, https://doi.org/10.1175/JAMC-D-18-0179.1.

In the abstract of your paper you say:

*Further, an additional bootstrap test revealed an underestimation in the warming rate varying from 0.5° to more than 4°C over Poland that was found to be largely influenced by the selection of few driving GCMs instead of considering the full range of possible climate model outlooks. Furthermore, we found that differences between various combinations of small subsets from the GCM ensemble of opportunities can be as large as the climate change signal.*

This certainly sounds like estimation uncertainty, due to the combination of a large spread in the ensemble and a small ensemble size. If it is, then if you applied the methods I describe in the manuscript to each subset, the differences between the subsets would be smaller, because the changes would all be closer to zero.

We also look at ways to evaluate downscaled results from large multi-model ensembles that involve 5 different levels. Two of these look at the ability of the downscaled GCM results reproduce the historical trends and interannual variability (e.g. Benestad, Rasmus; Parding, Kajsa; Isaksen, Ketil, Mezghani, Abdelkader (2016) "Climate change and projections for the Barents region: what is expected to change and what will stay the same?", ERL-102170.R2, DOI: 10.1088/1748-9326/11/5/054017). My question is how such efforts can be combined with SMMA/BMMA to improve our ability to assess the skill of the projections, e.g. for disaster modelling.

Probably the best way to influence the disaster risk modelling community is to publish results (as you have done) that then feed into the big reports (EU, IPCC). My experience has been that the summaries in the big reports are very influential (since many people don't have time to read all the individual papers).

What's the signal-to-noise ratio of the results in your paper (or equivalently, the p-value)? If they are clearly significant, then it wouldn't make sense to apply the methods described in our paper. If they are borderline significant, or not significant, then it might make sense.

The result in that paper that storminess in certain regions might increase is interesting. Is that a robust result across different studies, do you know, and is the change large enough to matter? If yes and yes then that's certainly something that the disaster risk management community should be interested in.

Best regards,

Steve Jewson